# A protein tertiary structure mimetic modulator of the Hippo signalling pathway

Hélène Adihou [1,2], Ranganath Gopalakrishnan [1,2], Tim Förster[2], Stéphanie M. Guéret [1,2], Raphael Gasper [3], Stefan Geschwindner[4], Carmen Carrillo García[5], Hacer Karatas[3], Ajaybabu V. Pobbati[6], Mercedes Vazquez-Chantada[7], Paul Davey[8], Carola M. Wassvik[1], Jeremy Kah Sheng Pang[9,10], Boon Seng Soh [9,10,11], Wanjin Hong [6], Elisabetta Chiarparin[8], Dennis Schade[5], Alleyn T. Plowright [1], Eric Valeur[1], Malin Lemurell [1], Tom N. Grossmann [12,13✉] & Herbert Waldmann [3,14✉]

Transcription factors are key protein effectors in the regulation of gene transcription, and in many cases their activity is regulated via a complex network of protein–protein interactions (PPI). The chemical modulation of transcription factor activity is a long-standing goal in drug discovery but hampered by the difficulties associated with the targeting of PPIs, in particular when extended and flat protein interfaces are involved. Peptidomimetics have been applied to inhibit PPIs, however with variable success, as for certain interfaces the mimicry of a single secondary structure element is insufficient to obtain high binding affinities. Here, we describe the design and characterization of a stabilized protein tertiary structure that acts as an inhibitor of the interaction between the transcription factor TEAD and its co-repressor VGL4, both playing a central role in the Hippo signalling pathway. Modification of the inhibitor with a cell-penetrating entity yielded a cell-permeable proteomimetic that activates cell proliferation via regulation of the Hippo pathway, highlighting the potential of protein tertiary structure mimetics as an emerging class of PPI modulators.

[1] Medicinal Chemistry, Research and Early Development Cardiovascular, Renal and Metabolism, BioPharmaceuticals R&D, AstraZeneca, Gothenburg, Sweden. [2] AstraZeneca-MPI Satellite Unit, Dortmund, Germany. [3] Max Planck Institute for Molecular Physiology, Dortmund, Germany. [4] Structure & Biophysics, Discovery Sciences, R&D, AstraZeneca, Gothenburg, Sweden. [5] Department of Pharmaceutical and Medicinal Chemistry, Christian-Albrechts-University of Kiel, Kiel, Germany. [6] Department of Multi-Modal Molecular (M3) Biology, A*STAR Institute of Molecular and Cell Biology, Singapore, Singapore. [7] Mechanistic Biology & Profiling, BioPharmaceuticals R&D, AstraZeneca, Cambridge, UK. [8] Medicinal Chemistry, Oncology R&D, AstraZeneca, Cambridge, UK. [9] Disease Modelling and Therapeutics Laboratory, A*STAR Institute of Molecular and Cell Biology, Singapore, Singapore. [10] Department of Biological Sciences, National University of Singapore, Singapore, Singapore. [11] Key Laboratory for Major Obstetric Diseases of Guangdong Province, The Third Affiliated Hospital of Guangzhou Medical University, Guangzhou, China. [12] Department of Chemistry and Pharmaceutical Sciences, VU University Amsterdam, Amsterdam, Netherlands. [13] Amsterdam Institute of Molecular and Life Sciences (AIMMS), VU University Amsterdam, Amsterdam, Netherlands. [14] Department of Chemistry and Chemical Biology, Technical University Dortmund, Dortmund, Germany. ✉email: t.n.grossmann@vu.nl; herbert.waldmann@mpi-dortmund.mpg.de

I n biomolecular assemblies, proteins adopt defined but flexible three-dimensional structures which are governed by numerous inter- and intramolecular amino acid contacts. The underlying protein–protein interactions (PPIs) critically depend on certain amino acids, so-called hotspots, which are usually presented by an arrangement of secondary structures at the surface of each binding partner[1]. Such binding epitopes have inspired the design of molecules that mimic protein secondary structures to reproduce the projection of hotspot amino acids[2,3]. To date, peptidomimetic approaches focus on the mimicry of β-hairpins[4], isolated loop structures[5], and α-helices[6,7] and have led to a number of bioactive inhibitors for biological targets non-addressable with small molecules. Noteworthy, there are proteins that resemble particularly challenging targets and show low targetability with peptidomimetic molecules. In these cases, more complex protein mimetics would be required to facilitate the spatial arrangement of multiple secondary structures. The feasibility of stabilizing such complex folds has been shown for multi-helix arrangements which required the use of multiple tailor-made intra-helical crosslinks, but those entities have not been used for the targeting of proteins so far[8–11].

Transcription factors represent an important class of particularly challenging protein targets as they are often embedded in complex regulatory PPI networks and lack defined binding pockets that would be suitable for small molecule targeting[12]. A prime example are the members of the transcriptional enhanced associate domain (TEAD) family which are downstream effectors of the Hippo signalling pathway and crucially involved in tissue homeostasis, cell proliferation, and organ growth control, such as in the heart[13]. This protein family is an attractive therapeutic target since promotion of TEAD activity and translation of downstream target genes were reported to stimulate regenerative processes[13,14]. One of TEAD's interaction partners is the co-repressor protein named transcription cofactor vestigial-like protein 4 (VGL4), which binds TEAD at a conserved site and inhibits its translational activity[15,16]. In biological experiments, it was shown that inhibition of the interaction between TEAD and VGL4 can initiate tissue repair and cardiac regeneration rendering an inhibitor of this interaction attractive for regenerative medicine[17]. Importantly, there are no inhibitors of this PPI available which prevents an exploration of this targeting strategy.

Here, we report the development of a proteomimetic molecule that stabilizes a two-helix arrangement derived from VGL4 designed to bind TEAD and to inhibit its interaction with VGL4. The inhibitor complex with TEAD was characterized using surface plasmon resonance (SPR) measurements and X-ray crystallography verifying the anticipated binding mode. Modification of this proteomimetic with a cell-penetrating entity yielded a bioactive inhibitor with robust cellular uptake and sufficient protease stability to perform cell-based experiments. Proximity ligation assays (PLAs), the analysis of target gene levels, and cell cycle alteration of cardiomyocytes verify TEAD engagement and the modulation of the Hippo pathway.

## Results

**Stabilized macrocyclic two-helix motif.** The TEAD protein is composed of two functional domains, a transcriptional enhancer factor domain to bind the target DNA and a transactivation domain which interacts with co-repressor VGL4 (ref. [18]) or co-activators such as the yes-associated protein (YAP) and the transcriptional co-activator with PDZ-binding motif (TAZ)[19]. For mouse TEAD4 (mTEAD4), a crystal structure has been reported in complex with the co-repressor VGL4(203–256) (**1**, Fig. 1a, PDB: 4LN0)[18]. mTEAD4 is highly homologous to human TEAD1 (hTEAD1, Supplementary Fig. 1), and mouse and human

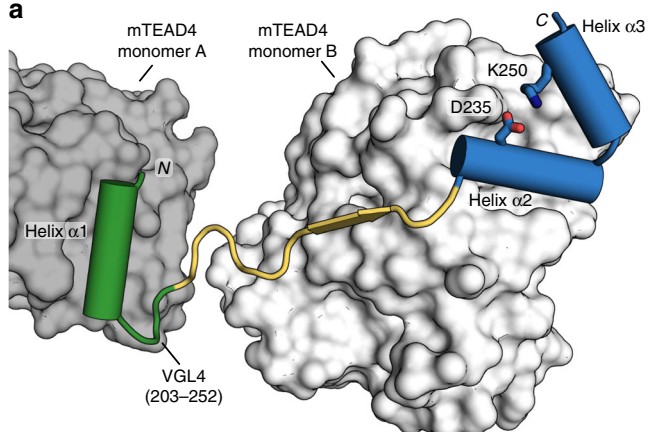

**Fig. 1 Structure-based design was applied to stabilize a two-helix motif.** **a** Crystal structure of the VGL4–TEAD (PDB: 4LN0)[18] protein–protein interaction. Helix α1 of VGL4 binds to TEAD monomer B while β-strand, helices α2 and α3 bind to TEAD monomer A. The two-helix motif of VGL4 shows potential for cyclization between D235 and K250. **b** Truncation study of VGL4. Fragments of the TEAD-binding motif of VGL4 were tested regarding their affinity for mTEAD4 and hTEAD1 using SPR ($K_d > 100$ μM: half-maximal signal was not reached at highest concentration $n = 1$ replicate). **c** Two-helix motif derived cyclic peptides obtained via lactamization of residues 235 and 250. Crosslink length-dependence of TEAD affinity was determined using mTEAD4 and hTEAD1 in an SPR assay (B: 2,4-diaminobutyric acid, O: ornithine, $n = 1$ replicate).

VGL4 have an identical TEAD binding sequence (Supplementary Fig. 2). This renders the mTEAD4–VGL4 crystal structure an excellent starting point for the design of ligands targeting hTEAD1.

The TEAD binding sequence of VGL4 can be divided into three fragments involving an N-terminal α-helix (**2**, helix α1, green) and a C-terminal two-helix motif (**4**, helices α2/α3, blue, Fig. 1a). Both sequences bind to the same area of two different TEAD transactivation domains (monomers A and B: dark and light grey, respectively). Peptide sequences **2** and **4** are connected by an extended stretch (**3**, yellow), which shows contacts with TEAD monomer A (light grey, Fig. 1a). The full-length VGL4-derived binding sequence **1** and the three isolated peptide motifs **2**–**4** were synthesized and their affinity for mTEAD4 and GST-hTEAD1 was determined by means of SPR measurements

(Fig. 1b). Among the four peptides, full binding sequence **1** shows the highest affinity for both TEAD variants ($K_d$ = 2.0/1.4 μM, mTEAD4/hTEAD1). Intervening peptide **3** exhibits very low affinity ($K_d > 100$ μM), which is in line with previously reported mutational studies indicating the lack of hotspot amino acids in this region[18]. Single-helix motif **2** ($K_d$ = 21.8/19.4 μM, mTEAD4/hTEAD1) shows moderate affinity while two-helix motif **4** ($K_d$ = 3.1/4.3 μM, mTEAD4/hTEAD1) is the highest affinity fragment in this panel (Fig. 1b). Sequence **4** was also reported to be essential for VGL4 binding to TEAD[18]. Notably, the isolated helices **5** and **6** derived from the two-helix motif **4** show low affinities for both TEAD proteins ($K_d > 100$ μM, Fig. 1b). Two-helix motif **4** was therefore selected as the starting point for structure-based design of a proteomimetic TEAD ligand.

When inspecting the structure of **4** bound to TEAD, we identified a salt bridge between K250 and D235 (Fig. 1a and Supplementary Fig. 3) that connects helices α2 and α3 and thus appears to stabilize the tertiary structure of **4** in its bound form. A variation of K250 to alanine results in tenfold reduced affinity ($K_d$ = 12/17 μM, mTEAD4/hTEAD1, Supplementary Table 1) indicating a contribution of the salt bridge to TEAD binding affinity. We reasoned that the replacement of this salt bridge by a covalent crosslink could pre-organize the bioactive conformation of the two-helix motif in solution and thereby increase the binding affinity for TEAD. For this purpose, different lactam bridges were incorporated ranging from five to eight crosslinking atoms (Fig. 1c). Macrocyclization via amide bond formation was performed on solid support after the full peptide sequence had been assembled via solid-phase peptide synthesis. The two amino acids for crosslinking were introduced with protecting groups cleavable under mild acidic conditions: carboxyl group (position X) with a 2-phenylisopropyl group and the primary amine (position Z) with a 4-methyltrityl substituent. After side chain deprotection, lactam formation was performed applying a double coupling with PyBOP and HOAt first for 3 h and then overnight. All macrocyclic peptides were obtained in good yields and purity. The two peptides with the shortest crosslink (**4A** and **4B**, 5 and 6 crosslinking atoms, respectively) show lower affinities than linear peptide **4**. For seven-atom crosslinks (**4C** and **4D**), moderate to good affinities were observed, depending on the location of the amide bond. For the longest crosslink (**4E**), highest affinities were obtained for both TEAD proteins ($K_d$ = 1.2 and 0.7 μM mTEAD4/hTEAD1) exceeding linear precursor **4** and the full-length VGL4 epitope **1** (Fig. 1c). Notably, the corresponding open versions of these macrocyclic peptides exhibit affinities in the range of peptide **4** (Supplementary Table 1).

## Structural characterization of stabilized two-helix motif 4E.

To understand the effects of macrocyclization in more detail, we investigated the secondary structure of the unbound peptides using circular dichroism (CD) spectroscopy. The CD spectrum of linear two-helix motif **4** indicates a relatively unstructured peptide with a helical content of only 10% (solid blue line, Fig. 2a). Notably, the two isolated helices (**5** and **6**) exhibit even lower helicity (both <5%, dashed lines). For macrocyclic peptide **4E** (orange), we observed considerably increased helicity (23%) indicating that macrocyclization and the resulting hairpin-like structure support helix formation.

To elucidate the binding mode of **4E** with mTEAD4, we pursued the crystallization of this complex resulting in crystals (space group P $2_1 2_1 2_1$) that provided a high-resolution crystal structure (1.3 Å, PDB: 6SBA). The overall structure of the complex between **4E** (orange) and mTEAD4 (grey) is very similar to the two-helix motif (blue) of VGL4 in its bound form (Fig. 2b) with the structures of mTEAD4 in both complexes superimposing

closely (RMSD = 0.949 Å, Supplementary Fig. 4). The electron density for **4E** (Fig. 2c) is well defined, clearly revealing the backbone conformation and orientation of the side chains. Notably, we also observed a defined electron density for the crosslink (Fig. 2c and Supplementary Fig. 5). **4E** (orange) targets the same binding site on TEAD as its unconstrained counterpart in VGL4 (blue, Fig. 2b). Also, both peptides exhibit a similar overall conformation (RMSD: 0.56 Å, Fig. 2d) thereby presenting analogous side chains towards the shallow hydrophobic groove on mTEAD4. This involves amino acids H237, F328, and L242 on helix α2 and T245, W246, and I249 on helix α3. The only noticeable difference between VGL4 and **4E** is a slightly larger angle between the two helices resulting in a minimal dislocation of the C-terminal part of helix α3 (max $\Delta l = 1.5$ Å). The orientation and contacts of the other side chains in **4E** are comparable to analogous residues in VGL4 (Fig. 2d). To elucidate the contribution of all amino acid side chains of **4E** to TEAD binding, an alanine scan was performed using SPR and a fluorescence polarization (FP) competition assay as readouts (Supplementary Table 3). Positions in which variation to alanine resulted in a considerable drop in affinity (fivefold increased $K_d$ value) are considered hotspot residues and are highlighted (red, Fig. 2e). Consistent with the crystal structure, we identified interface residues H237, F238, L242, W246, and I249. Surprisingly, the hotspot amino acid T245 is not located in the interface with mTEAD4 and variation to alanine may therefore influence the overall conformation of the free ligand.

## Cell-permeability and protease-stability of TEAD binding macrocycles.

Due to the intracellular localization of the TEAD–VGL4 complex, a potential inhibitor is required to penetrate cells efficiently. Cellular uptake of peptides was investigated using HeLa cells and flow cytometry as readout ($c$(peptide) = 5 μM, Supplementary Figs. 6 and 7). To minimize the load of extracellularly bound peptides, we applied a stringent washing protocol including tryptic digestion[20]. The cell-penetrating Tat peptide was used as reference[21,22] providing considerably increased fluorescence intensities (FI = 930) over background (FI = 95). However, peptides **4E** and **4** (FI = 228 and 245, respectively) show relatively low uptake, which we accounted to their negative overall charge (−1). Since it is known that negatively charged amino acids hamper cellular uptake and that overall positively charged peptides are more likely to penetrate cells[23,24], we considered the introduction of arginine residues and designed a set of peptides with increased overall charges (of up to +2, Supplementary Table 4). However, these modifications were not sufficient to promote meaningful uptake in HeLa cells as only small enhancements were observed (Supplementary Fig. 6). Therefore, N-terminal modification with the cell-penetrating Tat peptide was pursued providing peptide **7** (Fig. 3a), which shows unchanged affinity for both TEAD proteins (Supplementary Table 5) but very good cellular uptake by HeLa cells (FI = 18.300). Similar trends were also observed at 15 μM concentration (Supplementary Fig. 7).

With peptide **7** as a potential candidate for cell-based assays, we were interested to determine the stability and cell-permeability of its unlabelled analogue. Initially, the proteolytic stability in cell culture media containing foetal bovine serum (FBS) was investigated using HPLC/MS as readout. Importantly, about 80% of **7** (red) was intact after 24 h, indicating sufficient stability for cell-based assays. Notably, mass spectrometry analysis reveals that the degradation products harbour an intact macrocyclic core after 48 h (Supplementary Fig. 9). Along those lines, **4E** lacking the additional Tat sequence showed similar stability. Notably, a linear version of **4E** (**4E**(open)) exhibited about 60% degradation

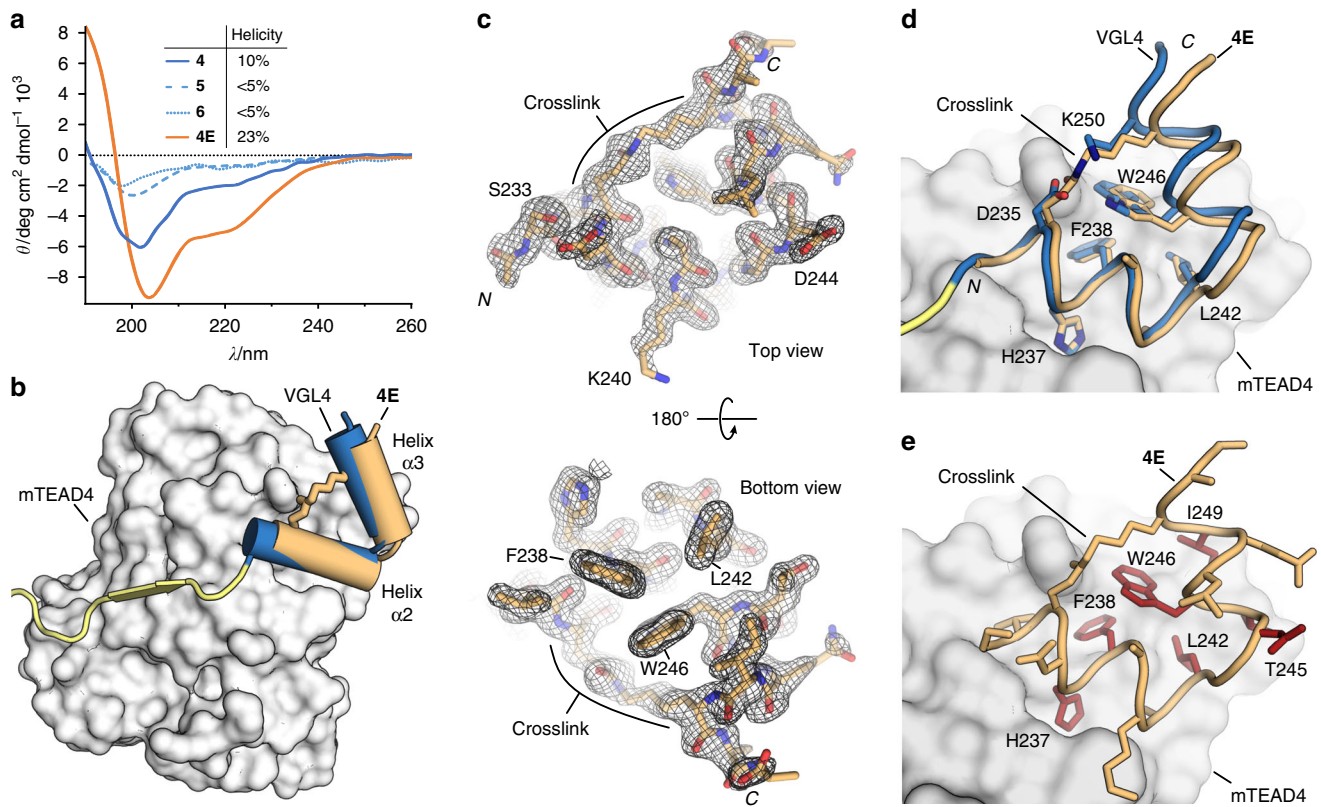

**Fig. 2 Structural characterization of the stabilized two-helix motif was performed. a** CD spectra of selected peptides and calculated helicities are shown (for details, see Methods section). Source data are provided as a Source Data file[42]. **b** X-ray crystal structure of **4E** (orange) co-crystallized with mTEAD4 (grey, PDB: 6SBA) superimposed with the corresponding part of the TEAD-binding motif of VGL4 (blue/yellow, PDB: 4LN0). **c** 2Fo-Fc electron density map (black, contoured at $\sigma = 1.2$) of **4E** (PDB: 6SBA) from top and bottom view. **d** Superimposition of two-helix motif from VGL4 (blue, PDB 4LN0) and peptide **4E** (orange, PDB: 6SBA), side chains of residues previously identified as interacting with TEAD are shown explicitly. **e** Structure of **4E** in complex with mTEAD4 (PDB: 6SBA) including all side chains is shown. Hotspot amino acids are highlighted (red) as determined by experimental alanine-scanning using SPR (mTEAD4) as readout. Crystal structure of the **E4**/mTEAD4 complex is available under PDB: 6SBA.

after 24 h (Fig. 3b) highlighting the protective effect that results from macrocyclization. To assess the cell penetration properties of unlabelled analogues, we determined their cytoplasmic and nuclear concentrations using a mass spectrometry-based methodology[25–27] and the Tat peptide as reference. RKO cells were incubated ($c$(peptide) = 25 μM) for 90 min and 24 h, then the cells were lysed and fractionated to isolate cytoplasmic and nuclear contents for subsequent UPLC-MS analysis. Linear peptide **4** and cyclized peptide **4E** accumulated in the cellular membrane (Supplementary Tables 10 and 11) and were not detected in subcellular fractions, unlike peptide **7** which was found in both compartments. We observed similar concentrations of **7** in the cytosol and nucleus after 90 min (Fig. 3c). After 24 h, we observed high intracellular concentrations suggesting sufficient biostability of peptide **7** for cell-based activity tests (Supplementary Fig. 12 and Supplementary Table 13). For the Tat peptide alone, overall higher uptake is observed with more pronounced accumulation in the nucleus indicating that the peptide cargo interferes with Tat cell penetration and distribution properties (Fig. 3c).

Next, the ability of the peptides in our panel to inhibit the TEAD–VGL4 interaction was investigated. For this purpose, a competition pull-down experiment was performed using biotinylated VGL4(203–256) (b-**1**) as the bait for GST-tagged hTEAD1. b-**1** was immobilized on streptavidin magnetic beads and hTEAD1 pull-down was detected using gel electrophoresis (Fig. 3d). As expected, peptides **4**, **7**, and **4E** compete with TEAD–VGL4 complex formation while cell-penetrating peptide

Tat alone does not show activity. Besides the repressor VGL4, co-activator YAP also binds to TEAD. To evaluate how inhibitor binding discriminates between these two binding partners, we performed a competitive binding assay based on FP. For that purpose, hTEAD1 was preincubated with fluorescently labelled versions of VGL4(203–256) or YAP(50–100). The concentration-dependent displacement of both binders by peptide **7** was then assessed and the half maximal inhibitory concentration ($IC_{50}$) values determined (Supplementary Fig. 15 and Supplementary Table 7). We observed a 16-fold lower $IC_{50}$ value for VGL4 ($IC_{50} = 5.46$ μM) than for YAP displacement ($IC_{50} = 87.7$ μM), indicating that peptide **7** exhibits a clear preference for a competition with VGL4. Notably, required concentrations of peptide **7** for VGL4 competition are higher in the pull-down assay (ca. 100 μM, Fig. 3d) than in the homogeneous FP-based competition assay (ca. 10 μM). This discrepancy between pull-down and homogeneous assay formats was already observed before[20] and presumably owes to the high local concentrations of immobilized bait protein (here, VGL4) in the pull-down assay which results in reduced target (here, hTEAD1) off-rates. Therefore, in the pull-down experiment, higher competitor concentrations are needed than in a homogeneous assay format.

**Inhibition of the VGL4–TEAD interaction in cell-based assays.** The Hippo signalling pathway controls several genes involved in cell growth, migration, and proliferation[28,29]. Expression of Hippo target genes is activated after complex formation between TEAD and a co-activator (e.g., YAP) resulting in the assembly of

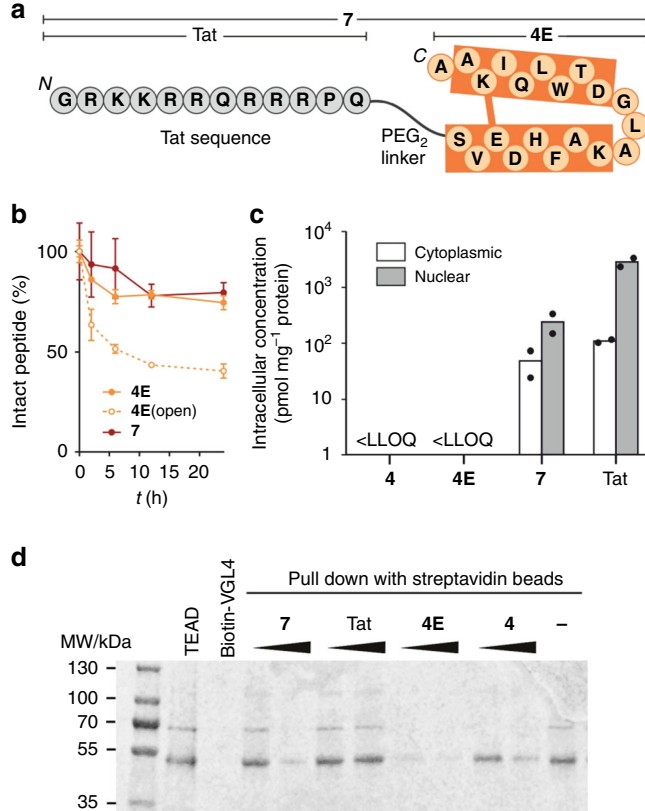

**Fig. 3 Cell permeability, protease stability, and PPI inhibition of modified peptides were investigated. a** Structure of peptide **7** consisting of peptide **4E** (orange) N-terminally linked via PEG$_2$ (8-amino-3,6-dioxaoctanoyl) to the cell permeable peptide Tat (grey). **b** Time-dependent proteolytic stability of the peptides in cell culture media buffer containing 10% foetal bovine serum ($c$(peptide) = 30 μM) ($n = 2$ technically independent replicates, error bars = SD). **c** Intracellular concentration of unlabelled peptides investigated by mass spectrometry (90 min incubation at $c$(peptide) = 25 μM). Peptides **4** and **4E** have concentrations below the lower limit of quantification (LLOQ), therefore they were not detected ($n = 2$ biologically independent replicates, error bars = SEM). **d** In vitro competition pull-down is performed with a complex of recombinant GST-hTEAD1 and two-helix motif peptides in presence of synthetic biotin-VGL4 immobilized on streptavidin magnetic beads; after incubation for 2.5 h, hTEAD1 pull-down is analysed by SDS-PAGE ($n = 1$ replicate). Source data are provided as a Source Data file.

the full transcriptional activator complex (Fig. 4a)[19]. Co-activators are regulated by phosphorylation-dependent degradation controlled via Hippo pathway effectors (e.g., SAV1, MST1/2, or LATS2)[30,31]. VGL4 on the other hand forms a repressor complex with TEAD blocking the transcription of Hippo target genes (Fig. 4a). Initially, we tested the influence of **7** on the YAP–TEAD interaction in a cellular context using a PLA in RKO cells. After 24 h incubation with **7**, PLA experiments show an increase in YAP–TEAD complexes, which is in line with an inhibition of the VGL4–TEAD interaction (Fig. 4b). Peptide **4E**, exhibiting lower cellular uptake, only shows a small change in PLA events while Tat has no significant effect. Increased YAP–TEAD complex formation and subsequent activation of Hippo target genes can be expected to induce cell migration and proliferation. For this reason, we performed a wound closure assay using RKO cells (Fig. 4c), which demonstrated no effect for **4E** and the Tat peptide while **7** shows considerably increased wound closure in a dose-dependent manner (Supplementary Fig. 16).

The proliferation promoting effects of the Hippo signalling pathway are crucial for heart muscle regeneration. Hippo pathway activity suppresses mitosis, and its inhibition reactivates cell cycle activity in cardiomyocytes in vitro and in vivo[17,32–34]. Given the ability of **7** to inhibit the formation of the VGL4–TEAD repressor complex, we tested its effect on cell cycle activity in postnatal rat cardiomyocytes. Therefore, we employed primary juvenile rat heart cells at postnatal days 5–6 (P5–P6) as these cells are on the verge of losing their proliferative capacity, entering a post-mitotic state[35]. Initially, cell permeability of **4E** and **7** was tested using flow cytometry, confirming robust cellular uptake of **7** in these primary heart-derived cells (Supplementary Fig. 17). In line with the MS-based cellular uptake experiments (Fig. 3c), we do not observe nuclear accumulation of peptide **7** being consistent with the low concentration of the nuclear target protein TEAD[36,37], which cannot be expected to alter the overall cellular distribution of peptide **7**. We then investigated the effects on the sub-cellular distribution of YAP in response to peptide treatment. Cardiomyocytes were treated with **7** for 4 h, and YAP localization analysed using YAP-specific antibodies and followed by fluorescent readout. Compared to the vehicle control, treatment with peptide **7** resulted in increased nuclear accumulation of YAP (Fig. 4d and Supplementary Fig. 18). A similar trend, though not as pronounced, was observed after 8 and 24 h (Supplementary Fig. 19). Having confirmed the nuclear localization of YAP, we then characterized the influence of **7** on the expression of TEAD target genes. Incubation of cardiomyocytes with **7** ($c = 30$ μM, $t = 18$ h) resulted in increased levels of the cell cycle genes *CYR61*, *CTGF*, *ANKRD1*, and *SEPINE1* (Fig. 4e and Supplementary Table 15) confirming the activation of these Hippo-associated genes. Notably, a cell cycle gene *CCNA2*, which has been shown to be insensitive to VGL4 levels, does not respond to peptide treatment. Also, the Tat control peptide does not show an effect in this assay (Fig. 4e). Functional efficacy of **4E** and **7** was measured as cardiomyocyte cell cycle activity after 4-days incubation in P5–P6 rat heart cells (Fig. 4f). The number of cycling cardiomyocytes was identified with an automated high-content imaging assay by means of Ki67/αActinin double-positive cells. Non-permeable peptide **4E** has no effect while **7** stimulates cell cycle activity to the same extent as positive controls SB (SB203580, p38MAPK inhibitor) and CHIR (CHIR99021, GSK3β inhibitor)[38–41]. This effect was further substantiated by a dose-dependent profile (Fig. 4g).

## Discussion

We report the structure-based design of a cell-permeable, macrocyclic proteomimetic that is derived from a two-helix motif originating from the transcriptional co-repressor VGL4 and was designed to bind the TEAD transcription factor. An X-ray crystal structure verifies that the proteomimetic binds TEAD in the anticipated manner, thereby having the potential to inhibit the TEAD–VGL4 interaction. Prior to cell-based assays, we confirmed efficient cellular uptake for the Tat-labelled proteomimetic (**7**) as well as sufficient stability in cell culture medium. We observed that incubation with compound **7** promotes the interaction between YAP and TEAD supporting the targeting of the TEAD repressor complex in RKO cells. In the same cell line, we also detected proproliferative effects of **7** confirming the expected activation of cell cycle activity. Also, proteomimetic **7** stimulates the expression of TEAD target genes in human cardiomyocytes as well as YAP nuclear translocation and cell cycle activity in primary juvenile rat heart cells which is a central feature required for cardiomyocyte proliferation. This proteomimetic is a rare example of a PPI inhibitor capable of activating transcription factor-mediated cellular events, and it highlights the potential of protein

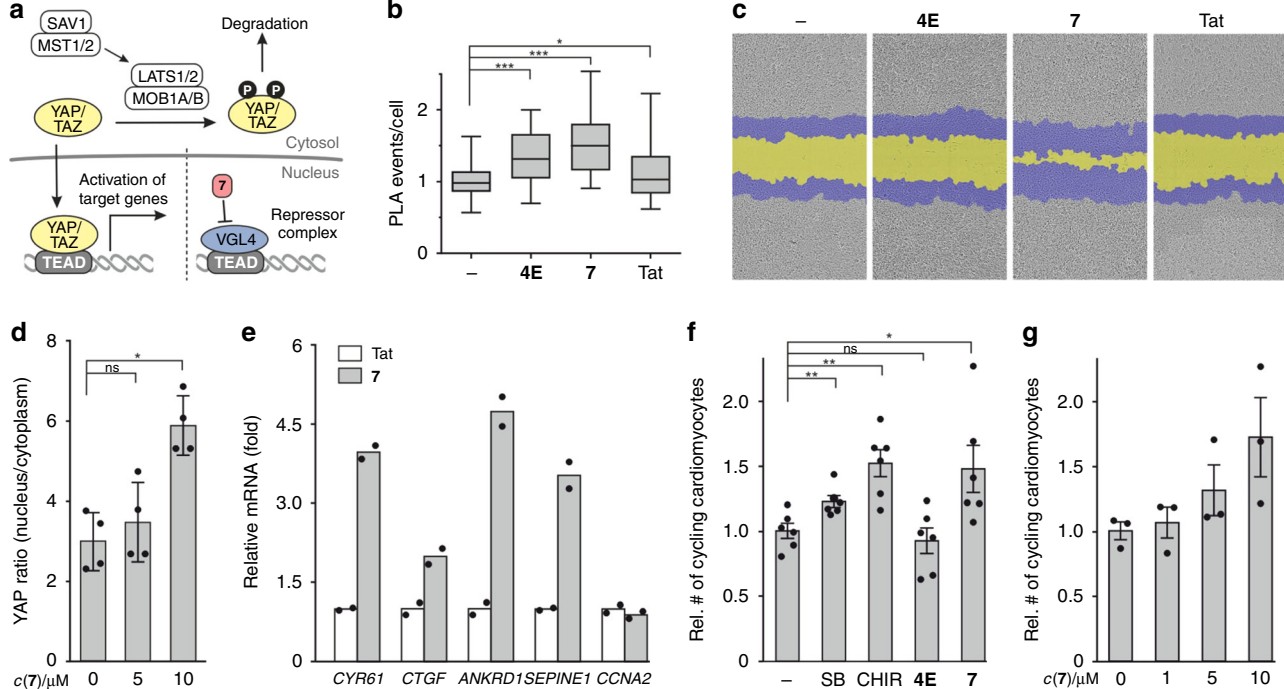

**Fig. 4 Cellular activity of 7 in RKO cells and in primary juvenile cardiomyocytes was investigated. a** Schematic overview of the Hippo signalling pathway. Presence of **7** inhibits formation of repressor complex, thereby supporting TEAD-dependent target gene expression. **b** Proximity ligation assay (PLA) to probe the YAP–TEAD interaction in RKO cells. Peptides were incubated for 24 h ($c = 30\,\mu M$). Untreated cells were used as control. PLA was performed using YAP and TEAD antibodies (centre: mean, whiskers: minimum to maximum, box: lower to upper quartile; for details, see Methods section, $n = 3$ biologically independent experiments with two $n = 2$ technically independent replicates each). **c** Representative images of wound healing assay of RKO cells treated with different peptides ($c = 30\,\mu M$, cell front is represented before (blue) and after 48 h treatment (yellow)). The experiment was performed in $n = 3$ biologically independent replicates each with $n = 3$ technically independent replicates. **d** Effects of **7** on YAP distribution (nucleus/cytosol). Juvenile rat heart cells were treated with peptide **7** (10 $\mu M$) or DMSO vehicle for 4 h, fixed and stained (DAPI, $\alpha$Actinin, YAP). Nuclear and cytosolic YAP levels in cardiomyocytes were quantified by image analysis (primary juvenile (P5–P6) rat cardiomyocytes, $n = 3$–4 biologically independent replicates, error bars = SEM). **e** qPCR analysis of the TEAD target genes. Bars represent the mRNA fold-changes of *CYR61*, *CTGF*, *SEPINE1*, *ANKRD1*, and *CCNA2* genes after 18 h treatment with **7** (red) relative to Tat (cardiomyocytes generated from human ES cell line (H7), $n = 2$ biological replicates with $n = 3$ technical replicates each). **f** High-content imaging-based analysis of cell cycle activity in primary juvenile (P5–P6) rat cardiomyocytes after 4 days. Untreated (–, DMSO) or treated ($c = 10\,\mu M$) with p38MAPK inhibitor SB203580 (SB), GSK3$\beta$ inhibitor CHIR99021 (CHIR), **4E** or **7** (plotted relative to DMSO-treated cells, 6 biological replicates, error bars = SEM). **g** Dose-dependent cell cycle stimulation after 4-day treatment with **7** (primary juvenile (P5–P6) rat cardiomyocytes, plotted relative to DMSO-treated cells, $n = 3$ biological replicates, error bars = SEM). For all the experiments, statistical significances were determined by Student's paired *t* test; $p < 0.05$ was considered statistically significant ($p < 0.05 = *$; $p < 0,01 = **$; $p < 0.001 = ***$, $p < 0.0001 = ****$) and source data are provided as a Source Data file.

tertiary structure mimetics as an emerging class of bioactive modalities.

## Methods

**General methods for peptides preparation**. Chemicals, including protected amino acids, were purchased from Merck, Roth, or Sigma-Aldrich and used without further purification. Solvents were purchased from Roth, Acros Organics, or VWR and used without further purification. The crude peptides were purified using Prep RP-HPLC (Agilent 1290 infinity hyphenated with 6120 quadruple Mass spectrometer) using Machrey Nagel C18 gravity, 5 μm, 125 mm/10 mm column. The purification was done with a flow rate of 20 mL min$^{-1}$, in a gradient of 10% B to 100% B in 45 min (buffer A: $H_2O$ + 0.1% TFA, B: MeCN + 0.1% TFA, detection at $\lambda = 214$, 220, 254, and 280 nm and MS detection). The purity of the peptide was checked by analytical RP-HPLC (Agilent 1290/1260 infinity HPLC) using Machrey Nagel column. The analysis was done at 1 mL min$^{-1}$, in a gradient of 5% B, 1% C to 65% B, 1% C in 30 min (A: $H_2O$, B: MeCN, C: $H_2O$ 90% / 10% TFA, detection at $\lambda = 214$, 220, 254 nm). HPLC-MS spectra were recorded on a LCQ Fleet from Thermo Ultimate 3000 series using a column a 50/2 Nucleodur C18 Gravity 1.8 μm with a flow rate of 0.4 mL min$^{-1}$ in a gradient of 0% B to 100% B in 10 min (A: $H_2O$ + 0.1% HCOOH, B: MeCN + 0.1% HCOOH). High-resolution mass spectra were recorded on a QLT Orbitrap mass spectrometer (with an electrospray ionization method) coupled to an Acceka HPLC using a Hypersyl GOLD. Peptide stock solutions were prepared in DMSO. For the acetylated peptide, concentration was determined using nanodrop with an extinction coefficient of 5690 M$^{-1}$ cm$^{-1}$. FITC-labelled peptides were measured at $\lambda = 495$ nm in 100

mM sodium phosphate buffer (pH 8.5) and calculated with an extinction coefficient of 77.000 M$^{-1}$ cm$^{-1}$ in JASCO V-550.

**Peptide synthesis**. Peptides were synthesized on a Syro I peptide synthesizer (Multisyntech) following standard Fmoc-protocols for solid-phase peptide synthesis. Peptide synthesis was performed on Rink amide LL AM/MBHA resin (Merck, 0.28 mmol g$^{-1}$, 0.05 mmol). To prepare the solutions, the amino acids were dissolved in a solution of oxyma pur 0.5 M in DMF, to obtain a final concentration of 0.5 M, HATU was dissolved in DMF to get a concentration of 0.5 M, DIPEA was dissolved in NMP to get a concentration of 2 M, piperidine was dissolved in DMF with a concentration of 25% and acetic anhydride was dissolved in NMP with a concentration of 10%. For the coupling, 400 μL of amino acids solution (0.2 mmol, 4 eq.) was mixed with 400 μL of HATU solution (0.2 mmol, 4 eq.) and 200 μL of DIPEA solution (0.4 mmol, 8 eq.) and added to the resin for 40 min. The amino acid couplings were followed by a capping of the remaining free amino groups with 800 μL of Ac$_2$O solution in presence of 200 μL of DIPEA solution for 2 min. Fmoc-deprotection was performed with 25% piperidine in DMF for 10 min. The subsequent aspartic acid and threonine residues in the sequence were introduced as a pseudoproline building block (Cas no. 920519-32-0). The resulting peptide resin was washed six times with 2 mL of DMF between each step. Each amino acid was coupled twice or thrice.

**Cyclization**. Aspartic acid and lysine were introduced, respectively, with 2-phenylisopropyl group and Mtt group as side chain protection. The 2-phenylisopropyl group and the Mtt group were removed by treating the resin with DCM/2% TFA/3% TIPS for 2 min ($9 \times 6$ mL). The peptide resin was then washed

with DCM (3 × 6 mL), 5% DIPEA in DCM (2 × 8 mL), NMP (3 × 6 mL), and DCM (3 × 6 mL). Then, 104 mg of PyBop (0.2 mmol, 4 eq.) and 28 mg of HOAt (0.2 mmol, 4 eq.) were dissolved in 3 mL of 1:1 NMP/DCM, 70 μL of DIPEA (0.4 mmol, 8 eq.) was then added. The mixture was added to the peptide resin and stirred for 3 h at room temperature, followed by a second coupling overnight. The peptide resin was washed with NMP (3 × 6 mL) and DCM (3 × 6 mL).

**Acetylation**. The peptide resin was deprotected using 2 mL of 25% piperidine in DMF (3 × 5 min) and then washed with DMF (3 × 6 mL) and DCM (3 × 6 mL). Then 134 μL of DIPEA and 120 μL of Ac$_2$O were dissolved in NMP, and the mixture was then added to the peptide resin. The suspension was stirred for 2 min at room temperature. The peptide resin was washed with DMF (3 × 6 mL) and DCM (3 × 6 mL).

**FITC labelling**. The N-terminal Fmoc group was deprotected using 2 mL of 25% piperidine in DMF (3 × 5 min) followed by washes with DMF (3 × 6 mL) and DCM (3 × 6 mL). Then 76 mg of Fmoc-O$_2$Oc-OH (0.2 mmol, 4 eq.), 76 mg of HATU (0.2 mmol, 4 eq.) were dissolved in 400 μL of DMF, followed by 70 μL of DIPEA (0.4 mmol, 8 eq.). The mixture was then added to the resin and stirred for 2 h. The peptide resin was washed with DMF (3 × 6 mL) and DCM (3 × 6 mL). The N-terminal Fmoc group was deprotected using 2 mL of 25% piperidine in DMF (3 × 5 min) followed by washes with DMF (3 × 6 mL) and DCM (3 × 6 mL). A solution of 0.10 g 5-Isothiocyanatofluorescein (0.250 mmol, 5 eq.) and 90 μL of DIPEA (0.5 mmol, 10 eq) in 800 μL DMF was prepared. The reaction mixture was shaken in the dark at room temperature for 3 h. The resin was washed with DMF (3 × 6 mL) and DCM (3 × 6 mL). The coupling procedure was repeated an additional time and this time shaken in the dark at room temperature for 16 h. The resulting bright yellow resin was washed with DMF (3 × 6 mL), DCM (3 × 6 mL) and dried under vacuum filtration.

**Tat/NLS labelling**. The N-terminal Fmoc group was deprotected using 2 mL of 25% piperidine in DMF (3 × 5 min) followed by washes with DMF (3 × 6 mL) and DCM (3 × 6 mL). Then 76 mg of Fmoc-O$_2$Oc-OH (0.2 mmol, 4 eq.), 76 mg of HATU (0.2 mmol, 4 eq.) were dissolved in 400 μL of DMF, followed by 70 μL of DIPEA (0.4 mmol, 8 eq.). The mixture was then added to the resin and stirred for 2 h. The peptide resin was washed with DMF (3 × 6 mL) and DCM (3 × 6 mL). The N-terminal Fmoc group was deprotected using 2 mL of 25% piperidine in DMF (3 × 5 min) followed by washes with DMF (3 × 6 mL) and DCM (3 × 6 mL). For coupling, amino acids (0.5 mmol, 10 eq.), 214 mg of COMU (0.5 mmol, 10 eq.), and 71 mg of Oxyma (0.5 mmol, 10 eq.) were dissolved in 1 mL of DMF. After the addition of 173 μL of DIPEA (1 mmol, 20 eq.), the solution was added to the peptide resin and the reaction mixture was shaken at room temperature for 20 min. The resin was washed with DMF (3 × 6 mL) and DCM (3 × 6 mL). For deprotection, the N-terminal Fmoc group was deprotected using 2 mL of 25% piperidine in DMF (3 × 5 min) followed by washes with DMF (3 × 6 mL), DCM (3 × 6 mL) and dried under vacuum filtration.

**Peptide cleavage and purification**. The peptidyl resin (0.05 mmol, 1 eq.) was cleaved with 1 mL of cleavage solution (triisopropylsilane 2.5%, water 2.5% in TFA) for 3 h at room temperature. The peptide was precipitated in 40 mL of cold diethyl ether/petroleum ether 1:1 after 10 min of centrifugation. The pellet was washed twice with 40 mL of diethyl ether. The residue was dried over vacuum, dissolved in water and lyophilized. The residue was dissolved in 1 mL of DMSO. The peptide was purified by reverse phase HPLC.

**Protein expression and purification**. CDNA of hTEAD1(209–426) fused with an N-terminal GST tag completed by a 6x His-tag and a 3C protease cleavage site was subcloned into pOPIN S (OPPF, University of Oxford, UK) vector. The resulting plasmid was transformed in *Escherichia coli* strain BL21 DE3 RIL(K+) (Agilent, Product #230280, Lot #0006370343) and the cells were grown in TB medium at 37 °C until the OD reached 0.6, then protein expression was induced using 400 μM of IPTG for 16 h at 18 °C. Bacteria were centrifugated and the pellet was resuspended in lysis buffer. Cell lysis was performed with a microfluidizer. After ultracentrifugation, the lysate was collected and subjected to purification on an ÄktaXpress system performing first an affinity chromatography separation on His Trap FF crude (GE Healthcare, USA) followed by size exclusion chromatography on SD75 26/60 (GE Healthcare, USA) using 20 mM HEPES, 100 mM NaCl, 1 mM TCEP, 2 mM MgCl$_2$, and 5% glycerol, pH 8. The purified protein was concentrated using Amicon Ultra Centrifugal Filters, 10K (Merck, Germany), aliquots were snap frozen and stored at −80 °C.

Mouse TEAD4(210–427) cDNA fused with a 6x His-tag and a 3C protease cleavage site was subcloned into pOPIN S (OPPF, University of Oxford, UK) vector. Plasmid was transformed in *E. coli* strain BL21 DE3 RIL(K+) (Agilent, Product #230280, Lot #0006370343) and the cells were grown at 37 °C in TB medium completed with lactose until the OD reached 0.6. Then the cells were incubated at 18 °C for 16 h. Bacteria were centrifugated and the pellet was resuspended in lysis buffer. Cell lysis was performed with a microfluidizer. After ultracentrifugation, the lysate was collected and subjected to purification on an ÄktaXpress system performing first an affinity chromatography separation on His

Trap FF crude (GE Healthcare, USA) followed by size exclusion chromatography on SD75 26/60 (GE Healthcare, USA) using 25 mM HEPES, 150 mM NaCl, and 1 mM TCEP, pH 7.2. The purified protein was concentrated using Amicon Ultra Centrifugal Filters, 10K (Merck, Germany), aliquots were snap frozen and stored at −80 °C.

**Surface plasmon resonance**. The SPR experiments were either performed on a Biacore S200 optical biosensor unit or a Biacore 8K optical biosensor unit (GE Healthcare). Sensor chips Series S CM5 (Research grade) were obtained from GE Healthcare. Prior to use, the sensor chips were equilibrated at room temperature for 15 min to prevent water condensation on the detector side of the sensor chip surface. A running buffer was prepared composed of 10 mM HEPES, 150 mM NaCl, and 0.05% (w/v) Tween-20, pH 7.4, and the system was equilibrated at 20 °C using a flow rate of 30 μL min$^{-1}$ after docking of the sensor chip. Ligand binding experiments have been performed applying the concept of multi-cycle kinetics. A contact time of 45 s was selected, followed by a 6-min dissociation phase to allow for complete dissociation of the analyte prior to the next cycle. The peptides have been dissolved in DMSO to a stock concentration of 10 mM. A digital dispenser (HP D300, Tecan) was used to dispense varying concentration of the ligands into running buffer provided in a standard 384-well plate and normalized with DMSO to 0.3% (v/v). Typically, seven concentrations of the analytes have been examined applying a threefold dilution pattern with 30 μM as top concentration. For the analysis, five running buffer blanks were injected to equilibrate the instrument. The data collection rate was set to 10 Hz, and all experiments have been repeated at least three times to allow for error estimations. The data have been analysed using Genedata Screener for SPR using the implemented steady-state data fitting routines and by applying a 1:1 binding model for the estimation of peptide affinities. Surface tethering of GST-hTEAD1 and mTEAD4 for SPR—for the covalent tethering of both GST–hTEAD1 and mTEAD4 onto the CM5 biosensor chip, running buffer at a flow-rate of 10 μL min$^{-1}$ was used. The carboxyl-dextran surface was activated for 7 min with 0.05 M NHS and 0.2 M EDC, followed by an injection of the proteins in 10 mM MES pH 6.4 at a concentration of 30–50 μg mL$^{-1}$. Contact times of 2–3 min were sufficient to achieve the desired densities of 2000–3000 RU. This was followed by a deactivation of the residual esters by injecting a solution of 0.5 M ethanolamine pH 8.0 for 7 min before engaging in ligand binding experiments. Reference surfaces were prepared accordingly, omitting the injection of protein over the activated reference surface.

**Circular dichroism**. Peptides were dissolved in 10 mM sodium phosphate pH 7.4 for a final concentration of 75 μM to 10 μM, depending on the peptide solubility. The spectra were obtained using a Jasco J-715 and a cuvette 0.1 mm. Represented CD spectra are an average of five measurements done between $\lambda = 190$ and 300 nm with a continuous scan mode, 50 nm min$^{-1}$ scan speed and data point every 1 nm. The α-helicity was calculated using CDNN software using the PEPFIT mode[42].

**Complexation of mTEAD4 with 4E and tag cleavage of mTEAD4**. mTEAD4 (4.5 g L$^{-1}$) was incubated with **4E** (2 eq.), then 3C protease was added and incubated over night at 4 °C. The cleaved protein complexed with Horseshoe was purified by size exclusion chromatography on SD75 26/60 (GE Healthcare, USA). The complex was concentrated using amicon ultra filter (Merck, Germany) to obtain a 1/1 mixture at 13 g L$^{-1}$.

**Crystallization of 4E/mTEAD4**. The purified complex was crystallized in a hanging drop vapour diffusion setup using 100 nL of complex (13 g L$^{-1}$) mixed with 100 nL of reservoir solution (0.1 M HEPES pH 7, 15% w/v PEG4000). Crystals appeared after 14 days and were harvested after 3 days. An initial dataset was taken in house on a Bruker Microstar generator with marDTB and mar345 image plate. The structure was solved with Phaser with chain A of PDB: 4LN0 as template. Iteration of manual model building with Coot and refinement with phenix.refine (including TLS refinement) improved the model to Rwork/Rfree of 20.4/23.6%. A second dataset taken at the Suisse Light Source X10SA beamline improved the resolution significantly. Rfree flags were transferred from the initial dataset using phenix and extended to 1.3 Å. Refinement was performed by phenix.refine.

**Fluorescence polarization**. Direct FP: the corresponding FITC-labelled peptide in DMSO was diluted with FP buffer (10 mM HEPES, 150 mM NaCl, 0.1% Tween-20, pH 7.4) to obtain a final concentration of 60 nM. mTEAD4/hTEAD1 was diluted with FP buffer in a twofold serial dilution in a 384-well plate. To 12.5 μL of the protein solution, 2.5 μL of the 60 nM peptide stock was added (final peptide concentration, 10 nM). After 1 h, FP was measured ($\lambda(ex) = 485$ nm; $\lambda(em) = 525$ nm). The dissociation constant ($K_d$) was determined from the binding curve with GraphPad from Prism.

Competition FP: N-terminally acetylated peptides were diluted 1:1 in a 384-well plate (5 μL, 75 μM−9 pM). Ten microlitres of a mixture (1:1) of mTEAD4 and FITC-labelled peptide was added (final concentrations: acetylated peptides = 50 μM−6 pM; mTEAD4 protein = 400 nM; FITC-labelled tracer peptide = 10 nM). After 1 h, FP was measured ($\lambda(ex) = 530$ nm; $\lambda(em) = 585$ nm). IC$_{50}$ was determined from the binding curve with GraphPad from Prism.

**HeLa cell culture for flow cytometry assay**. HeLa cells (DSMZ, #ACC-57) were purchased from DSMZ GmbH (Germany). HeLa cells were cultured at 37 °C with 5% $CO_2$ using DMEM (#P04-03500, PAN Biotech), supplemented with 10% (v/v) FBS, 1 mM sodium pyruvate (#P04-43100, PAN Biotech), 1% (v/v) non-essential amino acids (#P08-32100, PAN Biotech), 100 U mL$^{-1}$ penicillin, and 100 µg mL$^{-1}$ streptomycin (#P06-07100, PAN Biotech).

**Flow cytometry assay in HeLa cells**. Using 12-well plates, HeLa cells ($1 \times 10^5$ cells/well) were plated and allowed to grow in a growth medium for 24 h at 37 °C under humidified atmosphere with 5% $CO_2$. The cells were then treated with 5 µM and 15 µM final concentration of peptide in growth medium and a maximum of 0.5% DMSO. After 90 min of incubation at 37 °C under humidified atmosphere with 5% $CO_2$, the cells were washed with PBS three times. A solution of 0.05% trypsin and 0.02% EDTA in PBS (PAN Biotech) was added and the cells were incubated for 5 min at 37 °C under humidified atmosphere with 5% $CO_2$. Growth medium was used to stop the dissociation reaction and centrifugation (1.1 r.c.f., 3 min) allowed to harvest the cells. The cells were consecutively washed two times with PBS, suspended in 400–600 µL of PBS and transferred to a 5-mL Polystyrene round-bottom tube with cell-strainer cap (Falcon®, Corning Inc.). The analysis of the sample was recorded using a LSRII flow cytometer with forward and side scattered light to establish a gate for intact, live and non-aggregated cells. For each sample, a minimum of 10,000 events were collected in order to achieve a statistically relevant population. The fluorescence was set at 530 nm (filter: 530/30, mirror: 502 LP) to be in the wavelength excitation range of the FITC fluorophore (gating strategy is shown in Supplementary Fig. 8). The raw data were processed using FlowJo software (FlowJo version 10.1; source: Treestar, Inc.; http://docs.flowjo.com/d2/faq/general-faq/tree-star-flowjo/) and the results were shown as relative fluorescence intensities at 530 nm compared to DMSO-treated control and geometric means of fluorescence collected at 530 nm.

**Total intracellular and sub cellular peptide concentrations**. Peptides were incubated in RKO cells (ATCC, #CRL-2577) as follows: Experiment 1: RKO cells were incubated in 25 µM compound for 90 min and 24 h. Experiment 2: RKO cells were incubated in 25 µM for 30 min or 72 h. Following peptide incubation, the cells were washed twice with PBS, trypsinized and pelleted. The pellets were lysed in RIPA buffer (Perkin Elmer) for total lysis, or CelLytic NuCLEAR Extraction Kit (SIGMA) for subcellular fractionation. Total protein quantification BCA (Pierce) was performed prior to MS analysis to normalize sample data. Total and fractionated cell lysates were analysed by quantitative UPLC-MS utilizing a Waters Xevo TQ-XS (WBA0259) and an Acquity UPLC system from Waters consisting of Sample Manager (M16UFL953M), Acquity PDA (F17UPD457A), Column Oven (E17CMP703G) and Binary Solvent Manager (E17BUR621G). The Waters TQ-XS was operated in +ve ion Electrospray (ESI) mode with the optimized transitions for peptides **4**, **4E**, **7**, and TAT. The chromatograms at each transition were extracted, smoothed, and integrated to give the standard curve. Chromatograms of the samples were treated similarly, and by linear regression (1/x) an in-cell concentration was established in the re-suspended cell lysate using the Waters MassLynx TargetLynx$^{TM}$ product.

**Peptide stability in cell culture media**. A total of 30 µM of the peptide was incubated in MEM media supplemented with 10% FBS at 37 °C. At the indicated time point, 25 µL of the sample was withdrawn, mixed with an equal volume of the internal standard solution (6 µM Fmoc-Arg(Pbf)-OH), kept on ice for 15 min followed by centrifugation ($16 \times g$) at 4 °C for 5 min. The supernatant was analysed with LC-MS (UltiMate 3000 UHPLC Systems) using C18 reverse phase column (Nucleodur, 1.8 µm C18 gravity, 2 mm) and Thermo Orbitrap Velos Pro ETD. The peptide absorbance ($\lambda = 280$ nm) or the mass spectrometry peak intensity relative to that of the internal standard peak intensity was graphed. The stability experiments were performed in duplicates.

**Competition pull-down assay**. Synthetic biotinylated VGL4 was immobilized on 25 µL of streptavidin magnetic beads (#1420, New England Biolabs) for 1 h at room temperature. Beads were then extensively washed with the incubation buffer (300 mM NaCl, 25 mM Tris (pH 8.8), 2 mM DTT, 2% glycerol). Purified GST-hTEAD1 (1 µM) was incubated with the required amount of peptide (50 and 200 µM) in incubation buffer for 30 min at room temperature; then immobilized VGL4 was added to the protein/peptide complex (final volume 200 µL) and incubated for 2.5 h at room temperature. Beads were then extensively washed with the incubation buffer followed by protein elution after boiling the beads at 95 °C for 10 min in 25 µL of SDS loading buffer. SDS PAGE was performed using standard methods and the gel was visualized after Coomassie staining with a Li-COR Fc Imager.

**Primary culture of rodent newborn heart cells**. After dissociation, cardiac cells were homogeneously suspended in a medium and 4000 cells per well were seeded on a 0.1% gelatin-coated 384-well microclear plate (Greiner). Cells were incubated at 37 °C and 5% $CO_2$ in IMDM media supplemented with 50 µg mL$^{-1}$ penicillin/streptomycin, 0.1 mM non-essential amino acids, 0.1 mM β-mercaptoethanol (all from Gibco), and 2% FBS (Pan Biotech) for 6 days at 37 °C and 5% $CO_2$ (ref. [43]), unless otherwise was noted.

**Cell culture**. Rat experiments were approved by the local animal protection agency. Wistar rats were sacrificed on day 5–6 after birth (due to the low age of the rats, sex was not determined). The heart tissue was first minced and then gently dissociated by slowly pipetting up and down in the enzymatic solution provided in the dissociation kit (#130-098-373, Miltenyi Biotec). After three steps of mechanical dissociation, the cardiac cells were passed through a strainer to remove larger particles and the single-cell suspension was centrifuged (500 r.p.m., 15 min) and resuspended in growth media. The cells were then ready for the different applications.

RKO (ATCC, #CRL-2577) cells were purchased from ATCC (USA), HeLa cells were purchased from DSMZ GmbH (Germany). RKO cells were cultured at 37 °C with 5% $CO_2$ using Eagle's MEM (#P04-08500, PAN Biotech) containing 10% FBS (#10500-084, Invitrogen). HeLa cells were cultured at 37 °C with 5% $CO_2$ using DMEM (#P04-03500, PAN Biotech), supplemented with 10% (v/v) FBS, 1 mM sodium pyruvate (#P04-43100, PAN Biotech), 1% (v/v) non-essential amino acids (#P08-32100, PAN Biotech), 100 U mL$^{-1}$ penicillin, and 100 µg mL$^{-1}$ streptomycin (#P06-07100, PAN Biotech). Rat primary cardiomyocytes were incubated at 37 °C and 5% $CO_2$ in IMDM media supplemented with 50 µg mL$^{-1}$ penicillin/streptomycin, 0.1 mM non-essential amino acids, 0.1 mM β-mercaptoethanol (all from Gibco), and 2% FBS (Pan Biotech). The cells were regularly assayed for mycoplasma contamination using MycoAlert™ Mycoplasma Detection Kit (#LT07, Lonza) according to manufacturer's instructions.

**Proximity ligation assay**. Ca. 5000 RKO cells were seeded per well in black 96-well plates with glass bottom (#5241-20, MoBiTec) and treated with the peptides or DMSO as vehicle control. After 24 h, the cells were fixed for 10 min using 4% paraformaldehyde (#UN2209, AppliChem) in PBS prior to permeabilization with 0.1% Triton X-100 (#39795.02, Serva) in PBS for 10 min. Blocking was performed using 2% BSA (#11945.03, Serva) in PBS with 0.1% Tween-20 (#P2287, Sigma-Aldrich) for 1 h prior to incubation with the primary antibodies (mouse anti-YAP, #sc-101199, Santa Cruz, 1:100; rabbit anti-panTEAD, #13295, Cell Signalling, 1:1000) overnight. The cells were treated with the corresponding PLA probes (donkey anti-rabbit PLUS, #DUO92002, dilution 1:5; donkey anti-mouse MINUS, #DUO92004, dilution 1:5, Sigma-Aldrich) and with the Duolink In Situ Detection Orange Kit (#DUO92007, Sigma-Aldrich) for ligation and amplification according to the manufacturer's instructions. Finally, the cells were treated with 1 µg mL$^{-1}$ DAPI (#D9542, Sigma-Aldrich) to stain nuclei for further microscopy analysis. Images were acquired using a Zeiss Observer Z1 microscope with an 63× oil objective. Picture data were analysed by the ImageJ "analyze particles" macro to count PLA spots per cell. Ten pictures were counted for each condition per experiment. Statistical analysis was performed using Student's paired $t$ test. $P < 0.05$ was considered statistically significant ($p < 0.05 = *$; $p < 0,01 = **$; $p < 0.001 = ***$; $p < 0.0001 = ****$).

**Wound closure assay**. Ca. 100,000 RKO cells were seeded per well in a transparent ImageLock 96-well plate (#4379, Essen Bioscience) and treated with the samples the next day. Wound Healing Assay was performed using the IncuCyte® ZOOM system and the WoundMaker™ device (both Essen Bioscience). The scratch was performed according to manufacturer's protocol and images were acquired every 2 h. Image-based analysis of wound closure was conducted using the IncuCyte ZOOM 2016B software (Essen Bioscience). Confluency within the wound area was normalized to the confluency at the time of scratch preparation and plotted using GraphPad Prism 6.

**Quantitative PCR**. Cardiomyocytes were generated from human ES cell line (H7, #WA07, WiCell) through GiWi differentiation protocol[44]. At day 7, the medium was replaced with RPMI/B27 medium, supplemented with insulin (Miltenyi Biotec) for maturation and refreshed every 2 days. At day 15, the glucose level in the medium was slowly reduced to 0% and the cells were grown until day 21 to facilitate the selection of more matured cardiomyocytes that rely on β-oxidation. Tat or peptide **7** (30 µM) was added to cardiomyocytes and the cells were incubated with the peptides for 18 h. Total RNA was extracted using the Quick-RNA miniprep kit (Zymo Research) and its quality was assessed using the $A_{260}/A_{280}$ and $A_{260}/A_{230}$ ratios. cDNA synthesis was carried out using the high-capacity cDNA reverse transcription kit (Applied Biosystems) as per manufacturer's guidelines. Equal amounts of cDNA were used in the TaqMan gene expression assays to estimate the levels of YAP target genes, *ANKRD1* (Assay ID: Hs00173317_m1), *CTGF* (Assay ID: Hs00170014_m1), *CYR61* (Assay ID: Hs00155479_m1), and *SERPINE1* (Assay ID: – Hs00126604_m1) by qPCR. *GAPDH* (Assay ID: Hs02758991 _g1) was used as internal control. In the graph, the RQ Min and the RQ Max values of the technical triplicates are plotted. The experiment was done twice independently, and the results were consistent. Statistical significances were determined by Student's unpaired $t$ test. $P < 0.05$ was considered statistically significant ($p < 0.05 = *$; $p < 0,01 = **$; $p < 0.001 = ***$).

**YAP localization assay in juvenile rat cardiomyocytes**. We analysed the YAP sub-cellular distribution in cardiomyocytes by means of fluorescence intensity after immunocytochemistry. After 3 days in culture, the cardiac cells were treated and fixed 4, 8, and 24 h after treatment. The cells were stained using primary antibodies

against αActinin (A-7811, dilution 1:300, Sigma) and YAP (#14074, dilution 1:500, Cell Signalling). Secondary antibodies were used to detect the primary antibodies (goat anti-mouse Alexa Fluor 568, A-11004, dilution 1:1000 and goat anti-rabbit AlexaFluor 488, A-11008, dilution 1:1000, both from Thermo Fisher Scientific) and nuclei were counterstained with DAPI (2-(4-amidinophenyl)-1H-indole-6-carboxamidine, dilution 1:1000, Roth). The samples were then imaged and analysed on an ImageXpress XL automated microscope (Molecular Devices). Fluorescence integrated intensity of YAP in the nuclei and cytoplasm of cardiomyocytes was assessed and the ratios calculated. Data are depicted as means ± SEM. Statistical significances were determined by Student's paired $t$ test. $P < 0.05$ was considered statistically significant ($p < 0.05 = *$; $p < 0.01 = **$; $p < 0.001 = ***$).

**Cell cycle activity assay in juvenile rat cardiomyocytes.** Ca. 4000 cells were seeded per well on a 0.1% gelatin- (#P06-20410, Pan Biotech) coated 384-well μClear microtiter plate (#781906, Greiner). One day after seeding, the cells were treated with test compounds of interest. Treatments were applied with an ECHO 520 acoustic dispenser (Labcyte) and medium was added to the wells. Four days after treatment, the cells were fixed with 4% formaldehyde (Roth) and immunostained in 0.2% Triton-X100 (Roth) in PBS (Gibco) supplemented with 5% FBS. The cells were stained using primary antibodies against αActinin (dilution 1:300), a cardiomyocyte marker (Sigma, A-7811) and Ki67 (dilution 1:250), a cell cycle marker (abcam, ab16667). Secondary antibodies were used to detect the primary antibodies (goat anti-mouse Alexa Fluor 568, A-11004, dilution 1:1000, and goat anti-rabbit AlexaFluor 488, A-11008, dilution 1:1000, both from Thermo Fisher Scientific) and nuclei were counterstained with DAPI (2-(4-amidinophenyl)-1H-indole-6-carboxamidine, dilution 1:1000, Roth). Samples were then imaged and analysed on an ImageXpress XL automated microscope (Molecular Devices). Data are depicted as means ± SEM. Statistical significances were determined by Student's paired $t$ test. $P < 0.05$ was considered statistically significant ($p < 0.05 = *$; $p < 0.01 = **$; $p < 0.001 = ***$).

**Flow cytometry in juvenile rat cardiomyocytes.** For flow cytometric analyses, ca. 20,000 cells were seeded per well on a 0.1% gelatin- (Pan Biotech) coated 96-well plate (Sarstedt). The cells were incubated in the already described media at 37 °C and 5% $CO_2$. One day after seeding, the cells were manually treated once with the peptides of interest previously labelled with FITC. After 90 min treatment, the cells were trypsinized (trypsin 0.25%, Sigma) and thoroughly washed before the peptide uptake was analysed by FACS to avoid unspecific cell surface binding. Peptide incorporation in live cells was measured as means of FITC fluorescence intensity. Data are depicted as means ± SEM. Statistical significances were determined by Student's paired $t$ test. $P < 0.05$ was considered statistically significant ($p < 0.05 = *$; $p < 0.01 = **$; $p < 0.001 = ***$).

**Animal experiments statement.** All experiments were approved by the official State animal care and use committee (LANUV, Recklinghausen, Germany AZ 84_02.04.2014.A333) The mice are housed under specific-pathogen-free conditions according to the guidelines of the Federation for Laboratory Animal Science Associations (FELASA). All experiments were conducted in accordance with the German federal law regarding the protection of animals and 'Guide for the Care and Use of Laboratory Animals' (National Institutes of Health publication 8th Edition, 2011).

**Reporting summary.** Further information on research design is available in the Nature Research Reporting Summary linked to this article.

## Data availability

The materials and data reported in this study are available upon request from H.W. The crystal structure of the **E4**/mTEAD4 complex is available under PDB 6SBA. Source data are provided with this paper.

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

## Acknowledgements

We thank the Dortmund Protein facility for the assistance with protein expression and purification. We thank Sasikala Thavam (MPI Dortmund) for the scale-up and resynthesis of some peptides. We are grateful to Michael Schulz (MPI Dortmund) for his help with flow cytometry. We thank Loana Arns and Roland Roland Winter (TU Dortmund) for the help with circular dichroism experiments. We thank the beamline staff of S10XA at the SLS, Paul Scherrer Institute, Villigen, Switzerland, for their assistance and support. We thank Jan G. Hengstler and Georgia Günther (IfADo, Dortmund) for providing the animals used in this experiment. T.N.G. is grateful for support by the European Research Council (ERC; ERC starting grant, no. 678623).

## Author contributions

H.W., T.N.G., E.V., A.T.P., M.L., H.A., R.Go., and S.M.G. conceived and designed the project. H.A., R.Go., S.M.G., and H.K. performed the chemical synthesis. T.F., H.A., R.Go., and H.K. performed the biochemical experiments. T.F., R.Go., S.M.G., A.V.P., M.V.C., P.D., J.K.S.P., B.S.S., and C.C.G. performed the biological experiments. R.Ga., H.A., and S.G. performed the biophysical experiments. All the authors analysed the results. All authors discussed the results and commented on the manuscript. H.W., T.N.G. and H.A. prepared the manuscript.

## Funding

## Competing interests

The authors declare no competing interests.
