## [Peer Review File · Nature Communications]

Reviewers' comments:

Reviewer #1 (Remarks to the Author):

Hippo signaling pathway plays vital roles in development, tissue regeneration and tumorigenesis, and now is emerging as a druggable candidate for both cancer therapy and other research fields. The activities of the downstream transcription factors, mainly TEAD1-4, are regulated by their interactions with coactivators (YAP/TAZ) or corepressors (VGLL4); and disrupting the YAP-TEAD transcriptional complex formation has been previously shown to suppress the aberrant cancer cell proliferation and tumor formation. However, inhibitor of the TEAD-corepressor interaction has not been identified. Here in this manuscript, the authors designed a proteomimetic molecule based on the previously published TEAD-VGL4 crystal structure. This stabilized protein tertiary structure could disrupt the TEAD-VGL4 interaction, but promote the YAP-TEAD association. Through continuous optimization, the authors improved the stability, cell-permeability, and protease-stability of the peptidomimetic and tested its effect on cell migration and proliferation. The strategy presented in this work provides an example for the development of PPI inhibitors capable of activating transcription factor.

Overall, this is an interesting story with significant novelty to the hippo field and drug design. However, some mechanistic studies and in vivo experiments are missing, the author should provide more in vivo evidences to characterize the stimulating effects of their mimetics. I would recommend its publication in NC if the following issues were carefully addressed.

Major points

1. Previous study has already shown that TDU2 has much stronger interaction with TEAD4 when compared with TDU1. Figure 1b seems just like a verification. In figure 1c, the author modified the peptide via lactamization of residues 235 and 250. The corresponding residues were substitute to E, K, B or O for crosslinking. As my understanding, the residue cysteine was mostly used for cyclization. Why not using the common cysteine residue for crosslinking? Is there any preference of these amino acid substitutions?
2. In Figure 3c, though peptide 4, 7 and 4E can inhibit TEAD/VGLL4 complex formation in pull-down assays in vitro, Co-IP assay should be performed with or without the presence of these peptides. This may also examine the efficiency of the uptake by cells.
3. Currently, the evidence is insufficient to conclude that the peptidomimetics function through Hippo signaling. The authors showed that peptide 7 treatment promoted YAP-TEAD interaction using PLA assay. To firmly establish the activating effect, they should test this peptide in mice models as described in Ref. 15, 28 and 30 to examine whether this peptide can indeed promote cardiomyocyte proliferation, heart growth and cell cycle genes expression in vivo. This could be too much to ask, but it is important to prove that the peptide has any potential for practical application. I would not

insist on the animal study. However, the authors should at least, perform some cell-based assays to examine whether this peptide indeed induces YAP nuclear localization and target gene expression.

4. It seems that the competitive capacity of peptide 4E for inhibiting TEAD/VGL4 interaction is much higher than peptide 7 in Figure 3c. And 4E lacking the additional Tat sequence showed similar stability. Why did peptide 4E have no effect in the wound closure assay? Is this because of the delivery efficiency? Did the authors try other approaches to delivery it into cells?

5. As both VGLL4 and YAP bind to TEAD via the interface 2 (Cancer cell, 2014). Thus the peptide 7 might also inhibit YAP1-TEAD interaction. The author claimed peptide 7 can inhibit VGLL4-TEAD interaction but promote YAP/TEAD complex formation. Any theoretical explanation and experimental corroboration (except the PLA)?

Minor points

1. In figure 3c, molecular markers should be provided. The molecular weight of protein ladder should be labeled. Where is biotin labeled VGLL4 in this gel?
2. Half-life of a peptide is more useful for illustrating peptide stability.
3. Supplementary Figure 1 showed the sequence alignment of hTEAD1 and mTEAD4. However, the figure legend is "hTEAD4 and mTEAD4".
4. Page 3, line 6 from bottom, "Supplementary Table 2" should be "Supplementary Table 1".
5. The description in the manuscript should be consistent with the figure, e.g. "Sav1, Mst1, Mst2, Lats2" in the manuscript but "SAV1, MST1/2, LATS1/2" in Figure 4.
6. The Ramachandran statistics should be reported in Supplementary Table 2.

Reviewer #2 (Remarks to the Author):

This study describes the development of a peptide that inhibits the binding of transcription factor (TEAD) with its co-repressor (VGL4). The peptide comprises around 20 amino acids and was designed based on a two-helix region of VGL4. Two amino acids at both ends of the peptide – forming in VGL4 a salt-bridge - were connected by a covalent linker with the goal of stabilizing the VGL4 mimetic. The peptide binds TEAD with a K_d of 1.2 μ M. Conjugation of the peptide to the cell penetrating peptide TAT appears to increase the cellular uptake. The conjugate (compound 7) was tested in cell cultures and the authors report activation of cell proliferation via regulation of the Hippo pathway.

The strengths of this study are the sensible peptide inhibitor design approach based on mimicking a two helix region by a linker, and the thorough biophysical characterization of the various peptide variants by X-ray crystallography, alanine scanning, SPR and pulldown assay.

The weaknesses of the work are i) the rather weak binding affinity of the engineered peptides (e.g. peptide 4E, 1.2 μM) which appears not suited for inhibiting an intracellular target, ii) the moderate affinity improvement achieved with the engineering approach (less than 3-fold), iii) the high concentration of peptide needed to compete with the TEAD/VGL4 interaction (50 or 200 μM) which raises questions about the results in cells, and iv) the experiments in cells (cellular uptake, biological activity) which are not convincing to me. I describe the weak points in more detail below.

In my view, the developed peptide would need to be improved by a large factor (probably 100-fold or even more fold) to become a valuable tool, and the cell permeability and biological activity would need to be investigated in a greater depth to ensure that the effects observed are truly based on the disruption of the TEAD/VGL4 interaction by peptide 7, as claimed by the authors.

Major criticism:

1. The developed peptides (e.g. 4E) have a weak affinity. This means that micromolar concentrations would need to be reached in the cytosol to inhibit the interaction in cells. In fact, the pull down experiment showed that 200 μM of peptide 7 is required to compete with the interaction of TEAD/VGL4. In my opinion, the peptides with the weak affinity are not suited for experiments in cells and thus not suited as a tool at this stage. It is unlikely that someone would use the peptides in their current form as a tool.

2. The peptide design is sensible but the authors were unlucky that the affinity was not substantially improved by the covalent linkage (or cyclization). The affinity improvement achieved with the cyclization is less than 3-fold, which is disappointing considering all the effort with X-ray structure determination, alanine scanning, etc.

3. The stability of the cyclized peptide appears to be not much better than that of the linear one. While cyclization often yields a higher stability, the authors were unlucky in this case (as with the affinity). There are also concerns regarding the stability assay: i) The cyclic peptide (4E open) seems to plateau (Figure 3b), which is unexpected, and indicates a problem. ii) The stability was tested in culture media buffer containing 10% FBS, but it would be more relevant to test the stability in cell lysate.

4. Cell permeability: the authors have assessed the cell permeability by FACS which does not discriminate between localization in endosomes and the cytosol. It could well be that much of the peptide is trapped in the endosomes. One would need to use another assay, as for example a quantification of peptide via Halo tag, which is now used routinely to quantify peptide concentrations reached in the cytosol.

5. The results of the biological assays are not convincing because it is hard to believe that the peptide applied at 30 μM reaches a sufficiently high concentration in the cytosol (or nucleus) to interfere with the TEAD/VGL4 interaction. In fact, the peptide 7 showed only a weak effect in the pull down assay (in vitro) at concentration of 50 μM and no effect at 200 μM . A more rigorous experimental study is required. For example an analysis of the mRNA levels could tell if the peptide 7 follows the mechanism that the authors describe.

Reviewer #3 (Remarks to the Author):

The manuscript by Adihou et al. describes the design and characterization of a proteomimetic PPI inhibitor of the interaction between a transcription factor (TEAD) and its co-repressor, VGL4. For this purpose, by combining structure analyses and binding studies, the authors first identified a two-helix motif from the C-terminal part of VGL4 as a starting point for an inhibitor development. Given the proximity and interactions of the two helices, the author used a crosslinking strategy to stabilize synthetic mimetics. A number of macrocycles derived from the VGL4 C-terminus were generated and their binding to TEAD evaluated. The most efficient compound was further evaluated (CD, X-ray crystallography and alanine scan) and confirmed the benefit of the strategy. The compound was modified to allow cell-penetration and inhibition of the targeted TEAD/VGL4 interaction was evaluated in cell-based assays.

The article is particularly well written, pleasant to read, and the experiments clearly presented. The strategy, in line with the previous work of the authors, and the results are interesting both to

communities investigating the biological system targeted and to researchers involved in inter-domain research area.

Comments:

-SPR is extensively used in the study to extract KD values, however there is not a single sensorgram to evaluate quality of the measurements. Could the authors provide at least representative sensorgrams for the various experiments.

-There are discrepancies for the reported SPR values between hTEAD1 and compound 4E (0.7 uM in fig 1C, 1.3 uM in Supp Fig 7 and 1.13 uM in the legend of supp Fig 7).

-In the pull-down and SPR competition experiments, why was peptide 1 used to compete against hTEAD1-4E complex and not the opposite as would be expected for evaluation of an inhibitor? Could the experimental conditions for the SPR competition experiment be specified.

-my understanding is that YAP and the c-terminal part of VGL4 do not share the same binding site, hence the observed effect, could the authors discuss that aspect.

-Given the nature of compound 7, directly derived from VGL4, could it impact other partners of VGL4?

We would like to thank the reviewers for their thorough reading and suggestions. In response to the reviewer comments, we have performed additional experiments, e.g.

- MS-based quantification of intracellular peptide concentrations (Figure 3c, and additional SI data)
- fluorescence polarization-based TEAD competition assays (Supplementary Figure 14)
- analysis of effects on intracellular YAP distribution (Figure 4d)
- qPCR-based analysis of effects on Hippo target gene expression (Figure 4e)

Manuscript and supplementary information have been revised accordingly (changes highlighted in yellow). Below you can find a point-by-point response to all reviewer comments.

Reviewer #1 (Remarks to the Author):

Hippo signaling pathway plays vital roles in development, tissue regeneration and tumorigenesis, and now is emerging as a druggable candidate for both cancer therapy and other research fields. The activities of the downstream transcription factors, mainly TEAD1-4, are regulated by their interactions with coactivators (YAP/TAZ) or corepressors (VGLL4); and disrupting the YAP-TEAD transcriptional complex formation has been previously shown to suppress the aberrant cancer cell proliferation and tumor formation. However, inhibitor of the TEAD-corepressor interaction has not been identified. Here in this manuscript, the authors designed a proteomimetic molecule based on the previously published TEAD-VGL4 crystal structure. This stabilized protein tertiary structure could disrupt the TEAD-VGL4 interaction, but promote the YAP-TEAD association. Through continuous optimization, the authors improved the stability, cell-permeability, and protease-stability of the peptidomimetic and tested

its effect on cell migration and proliferation. The strategy presented in this work provides an example for the development of PPI inhibitors capable of activating transcription factor.

Overall, this is an interesting story with significant novelty to the hippo field and drug design.

However, some mechanistic studies and in vivo experiments are missing, the author should provide more in vivo evidences to characterize the stimulating effects of their mimetics. I would recommend its publication in NC if the following issues were carefully addressed.

Major points

1. Previous study has already shown that TDU2 has much stronger interaction with TEAD4 when compared with TDU1. Figure 1b seems just like a verification. In figure 1c, the author modified the peptide via lactamization of residues 235 and 250. The corresponding residues were substitute to E, K, B or O for crosslinking. As my understanding, the residue cysteine was mostly used for cyclization. Why not using the common cysteine residue for crosslinking? Is there any preference of these amino acid substitutions?

Response 1: In the published structure of VGL4, the distance between the two helices at the positions of macrocyclization (D235-K250) is about 9 Å (Figure 1a). Crosslink length variation showed that a minimum crosslink size of 8 atoms (Figure 1c) is needed. Introducing a disulfide via cysteine or homocysteine residues would result in crosslinks, not long enough to span the distance between the two helices. Therefore, we did not consider a disulfide bridge for the crosslink.

2. In Figure 3c, though peptide 4, 7 and 4E can inhibit TEAD/VGLL4 complex formation in pull-down assays in vitro, Co-IP assay should be performed with or without the presence of these peptides. This may also examine the efficiency of the uptake by cells.

Response 2: Attempts to perform Co-IP experiments in a cellular context failed due to the lack of an appropriate antibody. Alternatively, we confirmed the inhibitory effect of peptide **7** using *in vitro* competition assays based on fluorescence polarization (Supplementary Figure 14, Supplementary Table 7). In addition, the cellular uptake of non labelled peptides was determined (Figure 4e, Supplementary Figure 17, Supplementary Table 11, 12, 13 and 14). Using an MS-based assay, we determined the peptide concentration in the nuclear and cytoplasmic compartments and confirmed efficient cellular uptake of peptide **7**.

3. Currently, the evidence is insufficient to conclude that the peptidomimetics function through Hippo signaling. The authors showed that peptide **7** treatment promoted YAP-TEAD interaction using PLA assay. To firmly establish the activating effect, they should test this peptide in mice models as described in Ref. 15, 28 and 30 to examine whether this peptide can indeed promote cardiomyocyte proliferation, heart growth and cell cycle genes expression *in vivo*. This could be too much to ask, but it is important to prove that the peptide has any potential for practical application. I would not insist on the animal study. However, the authors should at least, perform some cell-based assays to examine whether this peptide indeed induces YAP nuclear localization and target gene expression.

Response 3: To further confirm the anticipated mode of action of peptide **7**, we investigated effects on the intracellular YAP distribution and target gene expression. We analysed YAP distribution (nucleus/cytosol) in cardiomyocytes using antibody labelling and fluorescence microscopy (Figure 4d). In presence of peptide **7**, YAP was translocated into the nucleus of rat cardiomyocytes which is in line with the expected mechanism of action. We then investigated the influence of peptide **7** on the level of TEAD target genes with a RT-qPCR experiment (Figure 4e) observing for peptide **7** the expected increased levels of these genes. Both experiments support the proposed mechanism of action for peptide **7** which involves the promotion of the YAP-TEAD complex, the nuclear translocation of YAP and the expression of TEAD target genes.

4. It seems that the competitive capacity of peptide **4E** for inhibiting TEAD/VGL4 interaction is much higher than peptide **7** in Figure 3c. And **4E** lacking the additional Tat sequence showed similar stability. Why did peptide **4E** have no effect in the wound closure assay? Is this because of the delivery efficiency? Did the authors try other approaches to delivery it into cells?

Response 4: **4E** does not show meaningful cellular uptake and can therefore not be expected to have an effect in cell-based assays including the wound closure assay. We did not test alternative strategies for cell delivery.

5. As both VGLL4 and YAP bind to TEAD via the interface 2 (Cancer cell, 2014). Thus, the peptide **7** might also inhibit YAP1-TEAD interaction. The author claimed peptide **7** can inhibit VGLL4-TEAD interaction but promote YAP/TEAD complex formation. Any theoretical explanation and experimental corroboration (except the PLA)?

Response 5: The general understanding of the protein-protein interaction network around TEAD and its cofactor is limited in particular with respect to the quantification of affinities and precise concentrations in the cellular context. However, to approach the reviewers question and assess the effect of peptide **7** on the VGL4/TEAD and YAP/TEAD complexes, we performed *in vitro* competition assays based on fluorescence polarization (Supplementary table 7). In line with our cell-based experiments, peptide **7** shows more pronounced competition with the VGL4/TEAD interaction ($IC_{50} = 5.5 \mu M$) than with the YAP/TEAD complex ($IC_{50} = 88 \mu M$, Supplementary Figure 14).

Minor points

1. In figure 3c, molecular markers should be provided. The molecular weight of protein ladder should be labeled. Where is biotin labeled VGL4 in this gel?
2. Half-life of a peptide is more useful for illustrating peptide stability.
3. Supplementary Figure 1 showed the sequence alignment of hTEAD1 and mTEAD4. However, the figure legend is “hTEAD4 and mTEAD4”.
4. Page 3, line 6 from bottom, “Supplementary Table 2” should be “Supplementary Table 1”.
5. The description in the manuscript should be consistent with the figure, e.g. “Sav1, Mst1, Mst2, Lats2” in the manuscript but “SAV1, MST1/2, LATS1/2” in Figure 4.
6. The Ramachandran statistics should be reported in Supplementary Table 2.

Response 7: The requested minor changes have been introduced.

Reviewer #2 (Remarks to the Author):

This study describes the development of a peptide that inhibits the binding of transcription factor (TEAD) with its co-repressor (VGL4). The peptide comprises around 20 amino acids and was designed based on a two-helix region of VGL4. Two amino acids at both ends of the peptide – forming in VGL4 a salt-bridge - were connected by a covalent linker with the goal of stabilizing the VGL4 mimetic. The peptide binds TEAD with a K_d of 1.2 μM . Conjugation of the peptide to the cell penetrating peptide TAT appears to increase the cellular uptake. The conjugate (compound 7) was tested in cell cultures and the authors report activation of cell proliferation via regulation of the Hippo pathway.

The strength of this study are the sensible peptide inhibitor design approach based on mimicking a two helix region by a linker, and the thorough biophysical characterization of the various peptide variants by X-ray crystallography, alanine scanning, SPR and pulldown assay.

The weaknesses of the work are i) the rather weak binding affinity of the engineered peptides (e.g. peptide 4E, 1.2 μM) which appears not suited for inhibiting an intracellular target, ii) the moderate affinity improvement achieved with the engineering approach (less than 3-fold), iii) the high concentration of peptide needed to compete with the TEAD/VGL4 interaction (50 or 200 μM) which raises questions about the results in cells, and iv) the experiments in cells (cellular uptake, biological activity) which are not convincing to me. I describe the weak points in more detail below.

In my view, the developed peptide would need to be improved by a large factor (probably 100-fold or even more fold) to become a valuable tool, and the cell permeability and biological activity would need to be investigated in a greater depth to ensure that the effects observed are truly based on the disruption of the TEAD/VGL4 interaction by peptide 7, as claimed by the authors.

Major criticism:

1. The developed peptides (e.g. 4E) have a weak affinity. This means that micromolar concentrations would need to be reached in the cytosol to inhibit the interaction in cells. In fact, the pull down experiment showed that 200 μM of peptide 7 is required to compete with the interaction of TEAD/VGL4. In my opinion, the peptides with the weak affinity are not suited for experiments in cells and thus not suited as a tool at this stage. It is unlikely that someone would use the peptides in their current form as a tool.

Response 8: Compound 7 shows robust cellular effects at 10 μM concentration. This is within a concentration range usually used for peptide-derived molecules that target intracellular protein-complexes. Importantly in this updated version of the manuscript, we have added a

number of experiments (e.g. YAP localization and TEAD target gene analysis) which verify the anticipated mode of action and prove the usefulness of compound **7**. Please note that in fluorescence polarization assays we observe nanomolar dissociation constants and that the high concentration of **7** in the biochemical competition assay results from the overall assay design.

2. The peptide design is sensible but the authors were unlucky that the affinity was not substantially improved by the covalent linkage (or cyclization). The affinity improvement achieved with the cyclization is less than 3-fold, which is disappointing considering all the effort with X-ray structure determination, alanine scanning, etc.

Response 9: We agree and in fact tried to optimize the peptide sequence but our efforts (also including not presented data) did not result in the desired affinity improvement. Nevertheless, we now clearly show that the affinity is sufficient to show bioactivity in this case.

3. The stability of the cyclized peptide appears to be not much better than that of the linear one. While cyclization often yields a higher stability, the authors were unlucky in this case (as with the affinity). There are also concerns regarding the stability assay: i) The cyclic peptide (**4E** open) seems to plateau (Figure 3b), which is unexpected, and indicates a problem. ii) The stability was tested in culture media buffer containing 10% FBS, but it would be more relevant to test the stability in cell lysate.

Response 10: In cell-based assays, the incubation with peptides is realized in cell culture medium containing 10% FBS. Therefore, we first assessed the proteolytic stability under these conditions. And here, macrocyclization providing peptide **4E** considerably increases protease resistance (Figure 3b). The addition of the Tat sequence in peptide **7** however introduces a new vulnerability. But importantly under our assay conditions the TEAD-binding core structure (**4E**) remains intact. As pointed out by the reviewer that does not reflect the situation inside the cell. However, using whole cell lysate, is in our case not an appropriate model either as it contains proteases from cellular compartments that should not be accessible for peptide **7**. Therefore, we decided to characterize the peptide stability using the in-cell concentration assay that verified both the cellular uptake and the presence of the intact inhibitor (Figure 3c). Here, we determined concentration of **7** after 90 min and 24 h incubation observing a loss of 30% for the total concentration, consistent with the culture media stability, and 47% for the cytosolic concentration whereas the nuclear concentration was relatively constant over time. This is particularly relevant as the targeted VGL4/TEAD PPI is located in the nucleus.

4. Cell permeability: the authors have assessed the cell permeability by FACS which does not discriminate between localization in endosomes and the cytosol. It could well be that much of the peptide is trapped in the endosomes. One would need to use another assay, as for example a quantification of peptide via Halo tag, which is now used routinely to quantify peptide concentrations reached in the cytosol.

Response 11: To address this question, we determined intracellular concentration of unlabeled peptides using a mass spectrometry-based assay (Figure 3c). Concentration of unlabeled peptides **4**, **4E**, **7** and Tat were determined in total cell but also in the cytosol and the nucleus after fractionation of the cell. Those results indicate that only peptide **7** and Tat can be found in both compartments with detectable concentrations.

5. The results of the biological assays are not convincing because it is hard to believe that the peptide applied at 30 μ M reaches a sufficiently high concentration in the cytosol (or nucleus) to interfere with the TEAD/VGL4 interaction. In fact, the peptide 7 showed only a weak effect in the pull down assay (in vitro) at concentration of 50 μ M and no effect at 200 μ M. A more rigor experimental study is required. For example an analysis of the mRNA levels could tell if the peptide 7 follows the mechanism that the authors describe.

Response 12: To address this point, we investigated the mRNA levels of specific TEAD target genes by RT-qPCR. We observed a significant increase of endogenous levels for CTGF, CYR61, SEPINE1 and ANKRD1 after 18 h of peptide 7 incubation ($c = 30 \mu$ M) in rat cardiomyocytes, indicating that the potency of peptide 7 is sufficient to induce a specific biological response. Please, also see Response 3 and 8 for more details.

Reviewer #3 (Remarks to the Author):

The manuscript by Adihou et al. describes the design and characterization of a proteomimetic PPI inhibitor of the interaction between a transcription factor (TEAD) and its co-repressor, VGL4. For this purpose, by combining structure analyses and binding studies, the authors first identified a two-helix motif from the C-terminal part of VGL4 as a starting point for an inhibitor development. Given the proximity and interactions of the two helices, the author used a crosslinking strategy to stabilize synthetic mimetics. A number of macrocycles derived from the VGL4 C-terminus were generated and their binding to TEAD evaluated. The most efficient compound was further evaluated (CD, X-ray crystallography and alanine scan) and confirmed the benefit of the strategy. The compound was modified to allow cell-penetration and inhibition of the targeted TEAD/VGL4 interaction was evaluated in cell-based assays.

The article is particularly well written, pleasant to read, and the experiments clearly presented. The strategy, in line with the previous work of the authors, and the results are interesting both to communities investigating the biological system targeted and to researchers involved in inter-domain research area.

Comments:

- SPR is extensively used in the study to extract KD values, however there is not a single sensorgram to evaluate quality of the measurements. Could the authors provide at least representative sensorgrams for the various experiments.

Response 13: The SPR sensorgrams of 4E, 4, YAP and VGL4 have been included in the supplementary information (Supplementary Figure 12 and 13).

- There are discrepancies for the reported SPR values between hTEAD1 and compound 4E (0.7 μ M in fig 1C, 1.3 μ M in Supp Fig 7 and 1.13 μ M in the legend of supp Fig 7).

Response 14: The values have been corrected.

- In the pull-down and SPR competition experiments, why was peptide 1 used to compete against hTEAD1-4E complex and not the opposite as would be expected for evaluation of an inhibitor? Could the experimental conditions for the SPR competition experiment be specified.

Response 15: Competing **4E** with peptide **1** in the SPR experiment was performed to confirm the reversibility of **4E** binding. To address the reviewer's question, we performed a homogeneous competition binding assay based on fluorescence polarization, and investigated the competition of peptide **7** with the YAP/TEAD and the VGL4/TEAD complex (Supplementary Figure 14). In line with the cell-based experiments, peptide **7** shows more pronounced competition with the VGL4/TEAD ($IC_{50} = 5.5 \mu M$) than with the YAP/TEAD complex ($IC_{50} = 88 \mu M$, Supplementary Figure 14).

- my understanding is that YAP and the c-terminal part of VGL4 do not share the same binding site, hence the observed effect, could the authors discuss that aspect.

Response 16: Indeed, the C-terminal domain of YAP is constituted by a beta-strand, an alpha-helix and a gamma-loop (PDB 3KYS). Although each structure element interacts with TEAD, the binding of YAP to TEAD is driven by the gamma-loop interacting with the interface 3 of TEAD. The binding mode of VGL4 is slightly different as it presents no loop but 2 helical regions (TD1 and TD2) and a beta-strand targeting the same interfaces than those secondary elements in YAP (PDB 4LN0, Figure 1a). Therefore, YAP and VGL4 only partially share a binding site on TEAD. YAP binding to TEAD is driven by the interaction formed by its loop at the interface whereas VGL4 binding to TEAD is controlled by the double-helix motif (Figure 1a and 1b).

- Given the nature of compound **7**, directly derived from VGL4, could it impact other partners of VGL4?

Response 17: To the best of our knowledge, myocyte enhancer factor 2 (MEF2) is the only other partner of VGL4 that potentially binds to the double helix motif which we used as our starting point (J. Biol. Chem. 2004, 279, 30800-6). However for the MEF2/VGL4 interaction, detailed structural characterization and affinity data is lacking.

REVIEWER COMMENTS

Reviewer #1 (Remarks to the Author):

This is a manuscript that I reviewed earlier. I had a chance to go through the rebuttal to my points and the authors have made several new experiments to address my major concerns. The revised manuscript is overall improved in peptide cellular uptake and related cellular experiments. But there are still some remaining questions to be addressed.

(1) The author failed to address my points about performing a Co-IP assay with or without the presence of these peptides (peptide 4, 7 and 4E). The author claimed no appropriate antibody. Actually, many paper including TEADs have used the antibody for Co-IP assay, such as Nat Cell Biol 19, 996–1002 (2017).

(2) I consider it better to provide the immunofluorescence images left behind the statistic figures in figure 4d and supplementary figure 17.

(3) In supplementary figure 16a, the figure is not clear and obvious to claim the cellular uptake. Please substitute with a clearer image and also stained with plasma membrane marker to mark the cell border.

(4) Peptides nuclear localization issues: since the author has the FITC-labeled peptide and claimed peptide binding to TEAD4, they could easily show nuclear localization of peptide and TEAD4 colocalization.

(5) The paper has some errors in typeface. Italics should be used for gene symbols.

Reviewer #2 (Remarks to the Author):

The authors have made a substantial effort to address several of the questions and concerns. While I still like very much the first part of the study (peptide engineering, X-ray structure), I am still not fully convinced about the second part.

My main concern remains that the effects of the peptide in cellular assays reported by the authors do not follow the anticipated pathway and mechanism. This concern stems from the finding that high concentration of peptide was needed to compete in vitro with the TEAD/VGL4 interaction. Inside cells, the concentration of peptide is most likely much lower. I am concerned that the observed effects (which are not crystal clear based on the extent of the effect) might be artifacts. The authors made an effort to measure the concentration of peptide in the nucleus but I think that the method used is not ideal to provide a clear result.

I had hoped that a comprehensive gene expression study would bring clarity and confirm the mechanism of the peptide, but only a rather small study was performed, and without the required controls. To make my remaining concerns more clear, I write comments to the authors' answers below (labeled as "COMMENT").

In summary, I can recommend publication of the peptide engineering part which looks very convincing to me (also alone), but not the part about the effect of the peptide in the various cellular assays.

Reviewer #2 (Remarks to the Author):

This study describes the development of a peptide that inhibits the binding of transcription factor (TEAD) with its co-repressor (VGL4). The peptide comprises around 20 amino acids and was designed based on a two-helix region of VGL4. Two amino acids at both ends of the peptide – forming in VGL4 a salt-bridge - were connected by a covalent linker with the goal of stabilizing the VGL4 mimetic. The peptide binds TEAD with a K_d of 1.2 μ M. Conjugation of the peptide to the cell penetrating peptide TAT appears to increase the cellular uptake. The conjugate (compound 7) was tested in cell cultures and the authors report activation of cell proliferation via regulation of the Hippo pathway.

The strength of this study are the sensible peptide inhibitor design approach based on mimicking a two helix region by a linker, and the thorough biophysical characterization of the various peptide variants by X-ray crystallography, alanine scanning, SPR and pulldown assay.

The weaknesses of the work are i) the rather weak binding affinity of the engineered peptides (e.g. peptide 4E, 1.2 μM) which appears not suited for inhibiting an intracellular target, ii) the moderate affinity improvement achieved with the engineering approach (less than 3-fold), iii) the high concentration of peptide needed to compete with the TEAD/VGL4 interaction (50 or 200 μM) which raises questions about the results in cells, and iv) the experiments in cells (cellular uptake, biological activity) which are not convincing to me. I describe the weak points in more detail below.

In my view, the developed peptide would need to be improved by a large factor (probably 100-fold or even more fold) to become a valuable tool, and the cell permeability and biological activity would need to be investigated in a greater depth to ensure that the effects observed are truly based on the disruption of the TEAD/VGL4 interaction by peptide 7, as claimed by the authors.

Major criticism:

1. The developed peptides (e.g. 4E) have a weak affinity. This means that micromolar concentrations would need to be reached in the cytosol to inhibit the interaction in cells. In fact, the pull down experiment showed that 200 μM of peptide 7 is required to compete with the interaction of TEAD/VGL4. In my opinion, the peptides with the weak affinity are not suited for experiments in cells and thus not suited as a tool at this stage. It is unlikely that someone would use the peptides in their current form as a tool.

Response 8: Compound 7 shows robust cellular effects at 10 μM concentration. This is within a concentration range usually used for peptide-derived molecules that target intracellular protein-complexes.

COMMENT: 10 μM is a reasonable concentration but the conc. in cells is usually much lower than that applied in the medium. There is a high chance that the conc. inside cells is not sufficient to have an effect, in particular considering the weak K_d (1 μM) and the low activity seen in the in vitro competition assay.

Importantly in this updated version of the manuscript, we have added a number of experiments (e.g. YAP localization and TEAD target gene analysis) which verify the anticipated mode of action and prove the usefulness of compound 7.

COMMENT: The gene analysis is not comprehensive and important controls are missing (see below)

Please note that in

fluorescence polarization assays we observe nanomolar dissociation constants and that the high concentration of 7 in the biochemical competition assay results from the overall assay design.

COMMENT: This last point would need to be explained better by the authors.

2. The peptide design is sensible but the authors were unlucky that the affinity was not substantially improved by the covalent linkage (or cyclization). The affinity improvement achieved with the cyclization is less than 3-fold, which is disappointing considering all the effort with X-ray structure determination, alanine scanning, etc.

Response 9: We agree and in fact tried to optimize the peptide sequence but our efforts (also including not presented data) did not result in the desired affinity improvement. Nevertheless, we now clearly show that the affinity is sufficient to show bioactivity in this case.

3. The stability of the cyclized peptide appears to be not much better than that of the linear one. While cyclization often yields a higher stability, the authors were unlucky in this case (as with the affinity). There are also concerns regarding the stability assay: i) The cyclic peptide (4E open) seems to plateau (Figure 3b), which is unexpected, and indicates a problem. ii) The stability was tested in culture media buffer containing 10% FBS, but it would be more relevant to test the stability in cell lysate.

Response 10: In cell-based assays, the incubation with peptides is realized in cell culture medium containing 10% FBS. Therefore, we first assessed the proteolytic stability under these conditions. And here, macrocyclization providing peptide 4E considerably increases protease resistance (Figure 3b). The addition of the Tat sequence in peptide 7 however introduces a

new vulnerability. But importantly under our assay conditions the TEAD-binding core structure (4E) remains intact. As pointed out by the reviewer that does not reflect the situation inside the cell. However, using whole cell lysate, is in our case not an appropriate model either as it contains proteases from cellular compartments that should not be accessible for peptide 7. Therefore, we decided to characterize the peptide stability using the in-cell concentration assay that verified both the cellular uptake and the presence of the intact inhibitor (Figure 3c). Here, we determined concentration of 7 after 90 min and 24 h incubation observing a loss of 30% for the total concentration, consistent with the culture media stability, and 47% for the cytosolic concentration whereas the nuclear concentration was relatively constant over time. This is particularly relevant as the targeted VGL4/TEAD PPI is located in the nucleus.

COMMENT: The authors do not discuss why the cyclic peptide (4E open) seems to plateau (Figure 3b). This is unexpected and could point to a problem.

4. Cell permeability: the authors have assessed the cell permeability by FACS which does not discriminate between localization in endosomes and the cytosol. It could well be that much of the peptide is trapped in the endosomes. One would need to use another assay, as for example a quantification of peptide via Halo tag, which is now used routinely to quantify peptide concentrations reached in the cytosol.

Response 11: To address this question, we determined intracellular concentration of unlabeled peptides using a mass spectrometry-based assay (Figure 3c). Concentration of unlabeled peptides 4, 4E, 7 and Tat were determined in total cell but also in the cytosol and the nucleus after fractionation of the cell. Those results indicate that only peptide 7 and Tat can be found in both compartments with detectable concentrations.

COMMENT: The method used to quantify the peptides "inside the cell" or "in the nucleus" is not ideal as the kits used to extract molecules of specific compartments are not perfect and usually co-purify components that are not in the compartment. Better methods should be applied here.

5. The results of the biological assays are not convincing because it is hard to believe that the peptide

applied at 30 μM reaches a sufficiently high concentration in the cytosol (or nucleus) to interfere with

the TEAD/VGL4 interaction. In fact, the peptide 7 showed only a weak effect in the pull down assay (in vitro) at concentration of 50 μM and no effect at 200 μM . A more rigor experimental study is required. For example an analysis of the mRNA levels could tell if the peptide 7 follows the mechanism that the authors describe.

Response 12: To address this point, we investigated the mRNA levels of specific TEAD target genes by RT-qPCR. We observed a significant increase of endogenous levels for CTGF, CYR61, SEPINE1 and ANKRD1 after 18 h of peptide 7 incubation ($c = 30 \mu\text{M}$) in rat cardiomyocytes, indicating that the potency of peptide 7 is sufficient to induce a specific biological response.

Please, also see Response 3 and 8 for more details.

COMMENT: The number of mRNAs analyzed is very small (four) and no controls were included (e.g. peptides other than TAT, other genes, etc.) One mRNA was used as internal control (GAPDH), I guess for normalization? SD are shown of technical replicates and not biological replicates. The raw data (before normalization to GAPDH) is not shown. The data of the biological repetition is not shown. A proper gene expression analysis with a chip would be very useful and could potentially eliminate all my doubts about the effects of the peptide in cells.

We would like to thank the reviewers for their thorough reading. Based on their suggestions, we have added more detailed explanations to the manuscript and additional data to Figure 3e (qPCR analysis). In addition, Supporting Figure 16 was modified and Supporting Figure 17 and Table 15 were added. Please find below a detailed response to the reviewer comments.

The reviewer comments are shown in black and **our responses are shown in blue**. Text we **added to the manuscript is highlighted yellow**. If required for the context, we also included our initial responses (first revision) which are in grey.

Reviewer #1 (Remarks to the Author):

This is a manuscript that I reviewed earlier. I had a chance to go through the rebuttal to my points, and the authors have made several new experiments to address my major concerns. The revised manuscript is overall improved in peptide cellular uptake and related cellular experiments. But there are still some remaining questions to be addressed.

(1) The author failed to address my points about performing a Co-IP assay with or without the presence of these peptides (peptide 4, 7 and 4E). The author claimed no appropriate antibody. Actually, many paper including TEADs have used the antibody for Co-IP assay, such as Nat Cell Biol 19, 996–1002 (2017).

Response 1: To further investigate the VGL4/TEAD interaction, we aimed to perform Co-IP experiments by immobilizing endogenous VGL4 to demonstrate the effect of peptide 7 on the native PPI. We, in fact, failed to find antibodies specific enough for that purpose. The successful Co-IP assays for this interaction found in the literature usually involved exogenously expressed tagged VGL4 and TEAD and the use of antibodies against the tags and not the actual proteins. Hence, they used a non-native system with artificially high protein concentrations. In above mentioned article (Nat Cell Biol 19, 996–1002 (2017)), the authors report a Co-IP assay with immobilized YAP and not VGL4. To address the question originally raised by the reviewer (validation of target engagement by peptide), we performed alternative experiments:
i) direct binding to purified human TEAD1 with fluorescence polarization and surface plasmon resonance
ii) competition of the hTEAD1/VGL4 interaction in pull-down and fluorescence polarization assays

(2) I consider it better to provide the immunofluorescence images left behind the statistic figures in Figure 4d and Supplementary Figure 17.

Response 2: We added the Supplementary Figure 17 with representative immunofluorescence images.

(3) In supplementary figure 16a, the figure is not clear and obvious to claim the cellular uptake. Please substitute with a clearer image and also stained with plasma membrane marker to mark the cell border.

Response 3: We have included figures with increased resolution clearly showing localization of fluorescein fluorescence (from FITC labelled peptides) within the cells. Our cultures show well-separated, individual cells, enabling their microscopic examination. In the past years, we have invested significant efforts into establishing membrane staining protocols for 2D-cultured cardiomyocytes. We noticed that typical membrane stains, such as WGA (wheat germ agglutinin) are frequently used to analyze single cells in cardiac tissue (3D) and do not work in 2D-cultured cardiomyocytes. As an alternative, we use the cytoplasmic cardiomyocyte marker α Actinin. The FITC signal clearly co-localizes with the cytoplasmic marker α Actinin indicating internal cellular localization. Please also note that in the manuscript, we report a total of three methods to verify cellular uptake of peptides: flow cytometry, MS-based detection, and the microscopy presented here.

(4) Peptides nuclear localization issues: since the author has the FITC-labeled peptide and claimed peptide binding to TEAD4, they could easily show nuclear localization of peptide and TEAD4 colocalization.

Response 4: Indeed, if a significant fraction of the internalized peptide pool would bind to TEAD, one could expect a nuclear localization of the peptide. However, only small amounts of TEAD are available, and we only expect a small fraction of internalized peptide to bind to TEAD, which would not be clearly visible in our microscopy setup. This is also supported by MS-based subcellular localization of active peptide 7 (Figure 3c), indicating similar distribution between nucleus and cytosol. We have included the following information to the manuscript to explain this aspect in more detail “In line with the MS-based cellular uptake experiments (Figure 3c), we do not observe nuclear accumulation of peptide 7 being consistent with the low concentration of the nuclear target protein TEAD^{37,38} which cannot be expected to alter the overall cellular distribution of peptide 7.”

Please also note that not necessarily all TEAD is bound and released by the peptide, as for a biological effect (activation of HIPPO target genes) small amounts of TEAD can be sufficient. See our Response 6 for more details.

(5) The paper has some errors in typeface. Italics should be used for gene symbols.

Response 5: Fixed.

Reviewer #2 (Remarks to the Author):

The authors have made a substantial effort to address several of the questions and concerns. While I still like very much the first part of the study (peptide engineering, X-ray structure), I am still not fully convinced about the second part.

My main concern remains that the effects of the peptide in cellular assays reported by the authors do not follow the anticipated pathway and mechanism. This concern stems from the finding that high concentration of peptide was needed to compete in vitro with the TEAD/VGL4 interaction. Inside cells, the concentration of peptide is most likely much lower. I am concerned that the observed effects (which are not crystal clear based on the extent of the effect) might be artifacts. The authors made an effort to measure the concentration of peptide in the nucleus but I think that the method used is not ideal to provide a clear result.

Response 6: We do not expect the accumulation of the peptide in the nucleus (please see our Response 4 for details). In Figure 4, we provide various biological data that clearly support the effect of peptide 7 on the HIPPO signaling pathway:

- i) Proximity ligation assay (PLA) shows increased colocalization of TEAD and YAP transcription factor (Figure 4b).
- ii) Fluorescence microscopy shows increased nuclear localization of YAP (a pre-requisite for expression of HIPPO target genes).
- iii) qPCR analysis shows increased level of HIPPO target genes

The reviewer raises concerns about the discrepancy between the peptide concentration applied in the biochemical competition pull-down experiment ($c = 50 - 200 \mu\text{M}$) and the concentrations at which the peptide shows effects in cell-based assays ($c = 10 - 30 \mu\text{M}$). Here, it is essential to note that the bait molecule (biotinylated VGL4 peptide) is immobilized in high density on beads, which generates very high local concentrations – For that reason, off-rates for TEAD1 can be expected to be very low which results in an increased apparent K_d . That requires then higher competitor concentrations than in a homogeneous assay format. In fact, in our homogeneous competition fluorescence polarization assay we can use lower peptide concentrations ($\text{IC}_{50} = 5.5 \mu\text{M}$, Supplementary Figure 14c) which is in line with the concentration range observed in cell based experiments. To clarify these aspects, we added the following considerations to the manuscript: “Notably, required concentrations of peptide 7 for VGL4 competition are higher in the pull-down assay (ca. $100 \mu\text{M}$, Figure 3d) than in the homogeneous fluorescence polarization-based competition assay (ca. $10 \mu\text{M}$). This discrepancy between pull-down and homogeneous assay formats was already observed before²¹ and presumably owes to the high local concentrations of immobilized bait protein (here, VGL4) in the pull-down assay which results in reduced target (here, hTEAD1) off-rates. Therefore in the pull-down experiment, higher competitor concentrations are needed than in a homogeneous assay format.”

In addition, it is important to note that intracellular concentrations below the IC_{50} of the biophysical competition can result in considerable cellular effects when activation of a biological target is approached. This is due to the fact that often small amounts of a signaling hub (such as transcriptional coactivators as TEAD) can already trigger pathway activation.

I had hoped that a comprehensive gene expression study would bring clarity and confirm the mechanism of the peptide, but only a rather small study was performed, and without the required

controls. To make my remaining concerns more clear, I write comments to the authors' answers below (labeled as "COMMENT").

Response 7: Analyzing the expression levels of few bona fide YAP/TAZ-target genes is commonly used to assess YAP activation. Please note that we have used *GAPDH* as a reference gene. Following the reviewer's suggestion we have not added another control gene (*CCNA2*) to assess if peptide 7 activates genes globally. We show that the levels of *CCNA2* are unaffected upon peptide 7 addition. We have added the data to Figure 4e and included the following sentences: "Notably, a cell cycle gene *CCNA2* which has been shown to be insensitive to VGLL4 levels does not respond to peptide treatment. Also the Tat control peptide does not show an effect in this assay (Figure 4e)."

Although we presented the technical replicates data for the qPCR experiments, the experiment was performed independently twice and is also mentioned in the methods section. The relative quantitation (Max and Min) are now provided in Supplementary Table 15. Regarding raw data before normalization, this is a delta Ct experiment and the software will not generate fold change values without using the internal control for normalization. It is important that we normalize in order to rule out that the changes we observe are not due to the differences in the cDNA amounts between samples. Definitely, global gene expression analysis using microarray or RNA-seq would provide a better picture, but we believe that a detailed analysis in that direction is beyond the scope of this study.

In summary, I can recommend publication of the peptide engineering part which looks very convincing to me (also alone), but not the part about the effect of the peptide in the various cellular assays.

Response 8: We thank the reviewer for highlighting the peptide engineering section of the manuscript and believe that together with the provided cell-based assays, they make a strong case for the TEAD-targeting abilities of the reported peptide.

Our old response: Compound 7 shows robust cellular effects at 10 μ M concentration. This is within a concentration range usually used for peptide-derived molecules that target intracellular protein-complexes. **COMMENT:** 10 μ M is a reasonable concentration but the conc. in cells is usually much lower than that applied in the medium. There is a high chance that the conc. inside cells is not sufficient to have an effect, in particular considering the weak Kd (1 μ M) and the low activity seen in the in vitro competition assay.

Please see Response 6 above.

Our old response : Importantly in this updated version of the manuscript, we have added a number of experiments (e.g. YAP localization and TEAD target gene analysis) which verify the anticipated mode of action and prove the usefulness of compound 7. **COMMENT:** The gene analysis is not comprehensive and important controls are missing (see below)

Please see Response 7 above.

Our old response : Please note that in fluorescence polarization assays we observe nanomolar dissociation constants and that the high concentration of 7 in the biochemical competition assay results from the overall assay design. **COMMENT:** This last point would need to be explained better by the authors.

Indeed, that has been fixed. Please see Response 6.

Our old response: In cell-based assays, the incubation with peptides is realized in cell culture medium containing 10% FBS. Therefore, we first assessed the proteolytic stability under these conditions. And here, macrocyclization providing peptide 4E considerably increases protease resistance (Figure 3b). The addition of the Tat sequence in peptide 7 however introduces a new vulnerability. But importantly under our assay conditions the TEAD-binding core structure (4E) remains intact. As pointed out by the reviewer that does not reflect the situation inside the cell. However, using whole cell lysate, is in our case not an appropriate model either as it contains proteases from cellular compartments that should not be accessible for peptide 7. Therefore, we decided to characterize the peptide stability using the in-cell concentration assay that verified both the cellular uptake and the presence of the intact inhibitor (Figure 3c). Here, we determined concentration of 7 after 90 min and 24 h incubation observing a loss of 30% for the total concentration, consistent with the culture media stability, and 47% for the cytosolic concentration whereas the nuclear concentration was relatively constant over time. This is particularly relevant as the targeted VGL4/TEAD PPI is located in the nucleus. **COMMENT:** The authors do not discuss why the cyclic peptide (4E open) seems to plateau (Figure 3b). This is unexpected and could point to a problem.

Response 9: The fact that peptide degradation in cell media appears to be reduced after 20 h, could be explained by degradation or other form of inactivation of some of the proteases in the medium. Please note that incubations were performed at 37 °C.

Our old response: To address this question, we determined intracellular concentration of unlabeled peptides using a mass spectrometry-based assay (Figure 3c). Concentration of unlabeled peptides 4, 4E, 7 and Tat were determined in total cell but also in the cytosol and the nucleus after fractionation of the cell. Those results indicate that only peptide 7 and Tat can be found in both compartments with detectable concentrations. **COMMENT:** The method used to quantify the peptides "inside the cell" or "in the nucleus" is not ideal as the kits used to extract molecules of specific compartments are not perfect and usually co-purify components that are not in the compartment. Better methods should be applied here.

Response 10: The used protocol and kits have been developed for the identification of small molecule/peptide ligands and have been tested rigorously previously (see manuscript references 26, 27 and 28). The quantification is specific to each peptide, as we use compound specific MS MRM transitions and the absolute quantitation relies on calibration curves using the compound standard: as a consequence the quantitation is accurate even in a mixture, we do not need purified compound. References:

26. McCoull, W. et al. Development of a novel B-cell lymphoma 6 (BCL6) PROTAC to provide insight into small molecule targeting of BCL6. *ACS Chem. Biol.* 13, 3131–3141 (2018).
27. Linnane, E. et al. Differential uptake, kinetics and mechanisms of intracellular trafficking of next-generation antisense oligonucleotides across human cancer cell lines. *Nucleic Acids Res.* 47, 4375–4392 (2019).
28. Cromm, P. M. et al. Lipidated stapled peptides targeting the acyl binding protein UNC119. *ChemBioChem* 20, 2987–2990 (2019).

Our old response: To address this point, we investigated the mRNA levels of specific TEAD target genes by RT-qPCR. We observed a significant increase of endogenous levels for CTGF, CYR61, SEPINE1 and ANKRD1 after 18 h of peptide 7 incubation ($c = 30 \mu\text{M}$) in rat cardiomyocytes, indicating that the potency of peptide 7 is sufficient to induce a specific biological response. Please, also see Response 3 and 8 for more details. **COMMENT:** The number of mRNAs analyzed is very small (four) and no controls were included (e.g. peptides other than TAT, other genes, etc.) One mRNA was used as internal control (GAPDH), I guess for normalization? SD are shown of technical replicates and not biological replicates. The raw data (before normalization to GAPDH) is not shown. The data of the biological repetition is not shown. A proper gene expression analysis with a chip would be very useful and could potentially eliminate all my doubts about the effects of the peptide in cells.

Please see Response 7 above.

REVIEWERS' COMMENTS:

Reviewer #2 (Remarks to the Author):

The authors do not provide additional results or insights to convince me more of the in vivo results in this study. I looked again at the in vivo data (proximity ligation experiment, nuclear localization of YAP, qPCR data) but stay with my opinion that it is not fully clear if the peptide 7 truly shows the effects via binding to the anticipated target. It could be, but it could equally well be that the observed "small" effects are not generated via the mechanism proposed. As indicated before, a comprehensive genomic analysis of cells treated with and without peptide 7 could deliver a clear picture and an answer to this question. Having said this, I like to repeat that the first part of the study (peptide engineering, X-ray structure), that I consider of high quality, alone is interesting and it might be worth to consider publishing only this part.

REVIEWERS' COMMENTS:

Reviewer #2 (Remarks to the Author):

The authors do not provide additional results or insights to convince me more of the in vivo results in this study. I looked again at the in vivo data (proximity ligation experiment, nuclear localization of YAP, qPCR data) but stay with my opinion that it is not fully clear if the peptide 7 truly shows the effects via binding to the anticipated target. It could be, but it could equally well be that the observed "small" effects are not generated via the mechanism proposed. As indicated before, a comprehensive genomic analysis of cells treated with and without peptide 7 could deliver a clear picture and an answer to this question. Having said this, I like to repeat that the first part of the study (peptide engineering, X-ray structure), that I consider of high quality, alone is interesting and it might be worth to consider publishing only this part.

We have down-toned our conclusions regarding the link between biological effect and the Hippo targeting of the peptidomimetic:

Last sentences in abstract were changed:

~~From: Modification with a cell-penetrating entity, yielded a cell-permeable and protease stable proteomimetic that activates cell proliferation via regulation of the Hippo pathway. For the first time, an inhibitor of the TEAD-VGL4 interaction is reported highlighting the potential of protein tertiary structure mimetics as an emerging class of PPI modulators.~~

To: Modification of the inhibitor with a cell-penetrating entity yielded a cell-permeable proteomimetic that activates cell proliferation via regulation of the Hippo pathway, highlighting the potential of protein tertiary structure mimetics as an emerging class of PPI modulators.

Last sentence of Results part has been **removed**: ~~“These findings show that 7 can penetrate primary rat heart cells and stimulate cell cycle progression of cardiomyocytes.”~~

In the Discussion the following sentences have been changed:

~~From: An X-ray crystal structure verifies that the proteomimetic not only binds TEAD in the anticipated manner but also selectively inhibits the TEAD-VGL4 interaction.~~

To: “An X-ray crystal structure verifies that the proteomimetic binds TEAD in the anticipated manner thereby having the potential to inhibit the TEAD-VGL4 interaction.”

~~From: Most notably, we observed that incubation with compound 7 promotes the YAP-TEAD interaction verifying the targeting of the TEAD repressor complex in RKO cells.~~

To: We observed that incubation with compound 7 promotes the interaction between YAP and TEAD supporting the targeting of the TEAD repressor complex in RKO cells.

~~From: This is the first reported inhibitor of the TEAD-VGL4 interaction and a rare example of a PPI inhibitor capable of activating transcription factor mediated cellular events. This highlights the potential of protein tertiary structure mimetics as an emerging class of bioactive modalities.~~

To: This proteomimetic is a rare example of a PPI inhibitor capable of activating transcription factor mediated cellular events, and it highlights the potential of protein tertiary structure mimetics as an emerging class of bioactive modalities.